# FieldFormer: Locality-Aware Transformers for Spatio-Temporal Modeling on Sparse Sensor Networks

## Abstract

Spatio-temporal sensor data in real-world systems is often sparse, noisy, and irregular, making it difficult to infer global structure from limited observations. Under extreme sparsity, we run into the limits of identifiability of latent system states, making latent field reconstruction fundamentally underconstrained. In such scenarios, multiple physically plausible fields may remain consistent with the same observations, requiring reconstruction models to rely heavily on inductive biases regarding locality, transport structure, and spatial regularity.

Under such sparsity regimes, reliable reconstruction becomes concentrated around the observational support induced by the sensor network, making sensor-space modeling a more identifiable objective than point-estimate global field recovery. We introduce FieldFormer, a mesh-free transformer architecture designed for locality-aware sensor-space modeling in persistent sensor networks. For each query, FieldFormer aggregates local evidence using a learnable velocity-scaled distance metric that adapts neighborhood geometry to heterogeneous spatio-temporal relationships. Neighborhoods are constructed as fixed maximal sparse contexts over nearby sensors and bounded temporal windows, while learned velocity-scaled offsets modulate token geometry within this context, enabling stable and scalable inference under extreme sparsity. A local transformer encoder integrates neighborhood information, while global consistency is modeled through coordinate-based neural field formulation.

We evaluate FieldFormer across five benchmarks spanning synthetic and real-world spatio-temporal systems, including anisotropic heat diffusion, shallow-water dynamics, atmospheric transport fields, and pollution monitoring datasets. Our results reveal that locality-aware reconstruction provides strong advantages in persistent sparse sensor networks where local domains of dependence remain observed, enabling FieldFormer to consistently perform competitively against state-of-the-art baselines on synthetic sensor-space prediction tasks, with state-of-the-art performance on real-world datasets under highly sparse and noisy sensing regimes.

**Code Repository** Our code repository is available at `https://anonymous.4open.science/r/fieldformer-C9E4/README.md`. The README file contains detailed instructions regarding the code base.

## 1 Introduction

Many real-world systems in environmental monitoring, atmospheric science, urban infrastructure, and geophysical sensing evolve over space and time according to underlying physical processes, often described through partial differential equations (PDEs). A general form of such systems is

$$\partial_t \mathbf{u}(\mathbf{x}, t) = \mathcal{F}\big(\mathbf{u}(\mathbf{x}, t), \nabla\mathbf{u}(\mathbf{x}, t), \Delta\mathbf{u}(\mathbf{x}, t), \ldots; \boldsymbol{\theta}\big), \tag{1}$$

where $\mathbf{u}(\mathbf{x}, t) \in \mathbb{R}^C$ denotes the evolving latent state and $\boldsymbol{\theta}$ represents unknown or partially known physical parameters such as diffusivity, transport velocity, or source terms. In practice, however, these systems are

rarely fully observed. Instead, measurements are obtained from a sparse set of sensors distributed over space, producing noisy and incomplete observations of the underlying process.

This mismatch between continuous physical dynamics and sparse sensing creates a fundamental challenge for spatio-temporal field reconstruction. Under realistic sensing constraints, large portions of the spatial domain remain unobserved, while sensors provide dense longitudinal traces only at a small number of fixed locations. As a result, reconstructing a globally consistent latent field from sparse measurements becomes fundamentally underconstrained. Multiple physically plausible latent fields may remain consistent with the same observations, particularly under extreme spatial sparsity, requiring reconstruction models to rely heavily on inductive biases regarding locality, transport structure, and spatial regularity.

To formalize this setting, we consider the discrete-time dynamical system induced by numerical integration of the PDE:

$$\mathbf{u}_{t+1} = \Phi_\theta(\mathbf{u}_t),$$

with sparse observations given by a measurement operator

$$\mathbf{y}_t = \mathcal{O}(\mathbf{u}_t),$$

where $\mathbf{y}_t$ corresponds to sensor measurements collected at a finite set of spatial locations. This induces a fundamental observability mismatch: the latent process evolves in a high-dimensional continuous state space, while observations lie in a much lower-dimensional sensor space determined by the sensing topology. Consequently, the mapping from latent states and physical parameters to observations becomes many-to-one under sparse sensing.

Recent results on identifiability under sparse observations (Norden et al., 2025; Wieland et al., 2021) further imply that distinct physical parameterizations may produce observationally equivalent sensor trajectories over finite horizons. That is, there may exist

$$\theta_1 \neq \theta_2$$

such that

$$\mathcal{O}(\Phi_{\theta_1}^t(\mathbf{u}_0)) = \mathcal{O}(\Phi_{\theta_2}^t(\mathbf{u}_0)),$$

despite inducing globally different latent fields. As a consequence, sparse sensing fundamentally limits the identifiability of the latent spatio-temporal field itself: multiple globally distinct reconstructions may remain fully consistent with the same sensor observations. Under such regimes, globally accurate reconstruction away from observed support cannot be guaranteed without introducing additional structural assumptions or inductive priors regarding the underlying data-generating process.

In this work, we argue that under extreme sparse sensing, reliable reconstruction becomes concentrated around the observational support induced by the sensor network, making *sensor-space modeling* a more identifiable objective than point-estimate global field recovery. This identifiability gap concerns single deterministic (point-estimate) field predictions; methods that explicitly represent uncertainty over multiple plausible completions are a complementary route rather than a counterexample. Sensor-space modeling refers to reconstruction objectives concentrated around the persistent sensing topology induced by deployed sensors. Rather than interpreting reconstruction as exact recovery of the latent physical state, we instead view it as the construction of a physically plausible surrogate field that remains consistent with sparse observations and local transport structure.

Motivated by this perspective, we introduce **FieldFormer**, a mesh-free transformer architecture designed for locality-aware sensor-space modeling in persistent sparse sensor networks. FieldFormer is built around the observation that many physical processes exhibit localized domains of dependence, where the evolution at a query location is primarily influenced by nearby spatial-temporal interactions. Instead of relying on global attention or fixed graph connectivity, FieldFormer performs reconstruction through adaptive local neighborhoods that dynamically organize sparse observations according to learned spatio-temporal geometry.

For each query location, FieldFormer aggregates local evidence using a learnable velocity-scaled metric: a spatio-temporal distance in which spatial and temporal offsets are rescaled before neighborhood construction. Neighborhoods are constructed as fixed maximal sparse contexts, meaning fixed-size candidate sets

drawn from nearby sensors and bounded temporal windows. Within this context, each token carries the observed value together with rescaled relative offsets from the query; we refer to this query-relative coordinate information as the token geometry. A local transformer encoder models relational structure over irregularly sampled observations, while global consistency is represented through a coordinate-based neural field formulation. When partial physical knowledge is available, additional structural consistency can be incorporated through differentiable PDE-based regularization.

Our formulation differs fundamentally from prior approaches to spatio-temporal reconstruction. Classical interpolation methods such as kriging and Gaussian processes rely on restrictive smoothness assumptions and scale poorly under large irregular datasets. Graph-based methods and GNN-PINN hybrids require fixed connectivity structures that do not adapt to evolving spatial dependencies, and can be replicated by transformer based approaches (Joshi, 2025). On the other hand, transformer-based imputation models and coordinate-based neural field methods often rely on globally regularized latent representations. Such approaches generalize to unseen sensors only when their implicit priors align with the underlying data-generating process, while failing to efficiently exploit localized domains of dependence under persistent sparse sensing. In contrast, FieldFormer explicitly aligns its inductive bias with the localized observability structure induced by sparse sensor networks.

We evaluate FieldFormer across five benchmarks spanning both synthetic and real-world spatio-temporal systems, including anisotropic heat diffusion, shallow-water dynamics, pollution dispersion simulation, atmospheric transport fields, and pollution monitoring datasets. Our experiments show that FieldFormer achieves strong performance for sparse sensor network imputation, where the goal is to reconstruct or forecast measurements over persistent sensing topologies from incomplete and noisy observations. At the same time, we observe that globally unconstrained field reconstruction under extreme sparsity exhibits highly variable behavior across datasets and reconstruction methods, with no single modeling paradigm consistently outperforming others. Different architectures succeed or fail depending on how their implicit inductive priors align with the underlying data-generating process, highlighting the fundamentally underconstrained nature of global latent field recovery under sparse sensing.

These findings suggest that sparse physical field reconstruction should be interpreted not as exact latent state recovery, but as the construction of observationally grounded surrogate fields under severe observability limitations. While extreme sparsity fundamentally limits reliable global reconstruction, accurate sensor-space imputation remains achievable and practically useful across a broad range of applications, including environmental and climate monitoring (Hart & Martinez, 2006), atmospheric sensing (NCEP, 2000), pollution tracking (Iyer et al., 2022; Bhardwaj et al., 2025; Concas et al., 2021), urban infrastructure management (Li et al., 2017; Bhardwaj et al., 2023), industrial IoT systems (Khalil et al., 2021), and scientific sensor deployments (Hart & Martinez, 2006) where measurements are spatially sparse but temporally dense.

Overall, our work contributes both a practical framework for sparse spatio-temporal reconstruction and a conceptual perspective on the limits of inference under sparse sensing. By connecting locality-aware modeling with observability structure, we argue that reconstruction quality in physical AI systems is fundamentally shaped not only by model architecture, but also by the information geometry induced by the sensing process itself.

## 2 Related Work

**Spatio-Temporal Imputation under Sparse Observability.** Classical methods such as Kriging (Cressie, 1990) and statistical models with spatio-temporal kernels (Wikle et al., 2019; De Luna & Genton, 2005) provide principled approaches for interpolation from sparse measurements, but they typically rely on stationarity, smoothness, and kernel assumptions that may not hold across heterogeneous physical systems. They also scale poorly, with cubic complexity in the number of observations, and do not explicitly account for governing dynamics during interpolation. Physics-based numerical solvers such as finite difference and finite element methods (LeVeque, 2007; Hughes, 2003) provide mechanistic structure, but require sufficiently specified physical models, boundary conditions, and parameters, which are often unavailable or non-identifiable from sparse sensor data alone.

Learning-based approaches have been developed to model more complex spatio-temporal dependencies. Message-passing recurrent neural networks (MPRNNs) (Iyer et al., 2022) use learned dynamics over sensor networks; graph-based approaches such as Graph WaveNet (Wu et al., 2019) and ST-Transformer (Xu et al., 2020) encode spatial relations through fixed or learned adjacency structures; and transformer models such as ImputeFormer (Nie et al., 2024) capture long-range temporal dependencies in structured multivariate time series. These methods are effective when the sensing topology and data-generating process align with their architectural priors, but they generally treat sparse sensing as a missing-data problem rather than as an observability-constrained physical inference problem. In contrast, our work explicitly studies reconstruction under severe sparse sensing, where the goal is not unconstrained global field recovery, but reliable sensor-space modeling and surrogate field construction under limited observational support.

Uncertainty-expressive methods provide a complementary direction for sparse reconstruction. Conditional score-based diffusion models such as CSDI (Tashiro et al., 2021) model distributions over missing time-series values, while neural process models (Garnelo et al., 2018) and their convolutional variants (Gordon et al., 2020; Foong et al., 2020) learn maps from context observations to predictive stochastic processes over functions. Deep ensembles (Lakshminarayanan et al., 2017) offer a simpler route to predictive uncertainty by aggregating independently trained predictors. These methods are especially relevant when multiple global completions remain plausible. Our work is complementary: we focus on locality-aware sensor-space modeling and evaluate point predictions under RMSE/MAE, where distributional methods must be reduced to a predictive summary such as a mean or median.

**Mesh-Free vs. Mesh-Dependent Spatio-Temporal Modeling.** Prior work on spatio-temporal reconstruction can be broadly divided into mesh-free and mesh-dependent approaches. Mesh-free methods (Tancik et al., 2020; Sitzmann et al., 2020; Raissi et al., 2019; Cressie, 1990; Hensman et al., 2013) operate directly on continuous coordinates, support arbitrary query locations, and naturally accommodate irregular sensor layouts and multi-resolution inference. These models are attractive for sparse sensing because they are not tied to a fixed grid or sensor graph. However, under extreme sparsity, continuous query capability does not by itself guarantee identifiable global reconstruction: predictions away from observational support are necessarily governed by the model's implicit priors.

Mesh-dependent methods (Nie et al., 2024; Iyer et al., 2022; Xu et al., 2020; Wu et al., 2019; Liu & Pyrcz, 2023; Liang et al., 2024; Santos et al., 2023; Zhao et al., 2024; Li et al., 2020a; Rahman et al., 2022; Li et al., 2017) assume a fixed spatial discretization, such as dense grids or static sensor graphs, and typically treat missing values as masked entries rather than absent spatial observations. Such methods can learn strong global or low-rank priors over fixed domains, and may generalize well when these priors match the underlying data-generating process. However, they are less naturally suited to irregular, persistent sparse sensor networks where reliable prediction is concentrated around the deployed sensing topology. FieldFormer follows the mesh-free paradigm, but uses it for locality-aware sensor-space modeling rather than claiming unconstrained global recovery under sparse observability.

**Physics-Informed Neural Networks and Differentiable PDE Solvers.** Physics-Informed Neural Networks (PINNs) (Raissi et al., 2019) embed PDE constraints into neural network training by computing differential operators on model outputs via automatic differentiation and penalizing residual violations. This enables both forward and inverse modeling when the governing equations and sufficient observations are available. However, PINNs typically rely on global coordinate function approximators and dense collocation over structured domains. As noted by (Krishnapriyan et al., 2021), vanilla global PINNs often struggle in moderately complex PDE regimes and real-world scenarios. In sparse longitudinal sensing settings, where observations are dense in time but sparse in space, global PDE residual minimization may be weakly constrained or even misleading when parameters, forcing terms, and boundary conditions are unknown.

Other physics-integrated approaches, such as DiffTaichi (Hu et al., 2019) and JAX-FDM (Kochkov et al., 2021), enable differentiable programming over simulation pipelines, but still rely on mesh-based discretizations and sufficiently specified simulators. These assumptions limit their applicability when the available data consist of scattered sensor measurements and the underlying physical process is only partially known. FieldFormer instead incorporates physical structure as optional local regularization while prioritizing observational consistency over persistent sparse sensor networks.

**Transformers for Scientific Modeling and Field Inference.** Transformer architectures have increasingly been applied to scientific modeling tasks, including spatio-temporal prediction and PDE-informed learning. TransFlowNet (Wang et al., 2022) introduces physics-constrained loss terms into transformer models for spatio-temporal flow prediction, while (Lorsung et al., 2024) integrate attention-based sequence modeling with physics-informed PDE tokens. These approaches typically assume dense spatial coverage, complete or well-specified governing physics, or grid-aligned discretizations, and often rely on global attention mechanisms. Such assumptions are difficult to satisfy in sparse sensing regimes with unobserved forcing, uncertain parameters, and incomplete boundary information.

In contrast, FieldFormer restricts attention to adaptive local neighborhoods while remaining globally grounded through continuous coordinates. This design reflects the view that, under sparse observability, reliable reconstruction depends on whether the sensor network sufficiently covers local domains of dependence. Rather than using transformers to model an unconstrained global latent field, FieldFormer uses local attention to organize sparse observations according to learned spatio-temporal geometry.

**Hybrid Approaches for Learning with Physical Structure.** Recent work has explored hybrid approaches that combine learning with physical structure without explicitly solving governing PDEs. Neural Operators such as Fourier Neural Operators (FNOs) (Li et al., 2020a) and Graph Neural Operators (Li et al., 2020b) learn mappings between function spaces, but typically require dense grid-based inputs and outputs. This makes them less suitable for scattered sensor measurements or persistent sparse sensing regimes where large portions of the domain are never observed. Other hybrids, including Physics-Informed Neural Fields (Chu et al., 2022) and Koopman operator methods (Lusch et al., 2018), embed physical constraints or dynamical priors into latent representations, but generally assume dense trajectories, simulation-generated data, or sufficient observational coverage.

These approaches demonstrate the value of combining learning with physical structure, but they do not directly address the observability limits induced by sparse sensor networks. FieldFormer complements this line of work by focusing on locality-aware reconstruction under sparse observational support, where global field recovery is underconstrained and prediction quality depends critically on the alignment between model inductive bias and sensing topology.

## 3 Problem Formulation

**Global Field Recovery.** The most general reconstruction objective is to estimate the latent field over a global spatio-temporal query domain

$$\mathcal{Q} \subset \Omega \times [0, T],$$

where $\Omega$ denotes the full spatial domain. In this setting, a model seeks to learn a function

$$\hat{\mathbf{u}} : \mathcal{Q} \to \mathbb{R}^q$$

such that

$$\hat{\mathbf{u}}(\mathbf{x}, t) \approx \mathbf{u}(\mathbf{x}, t), \qquad (\mathbf{x}, t) \in \mathcal{Q}.$$

The corresponding global reconstruction objective can be written as

$$\min_{\phi} \sum_{(\mathbf{x}, t) \in \mathcal{Q}} \|f_\phi(\mathbf{x}, t; \mathcal{D}_\mathcal{S}) - \mathbf{u}(\mathbf{x}, t)\|_2^2 + \lambda \mathcal{R}_{\text{phys}}(f_\phi),$$

where $f_\phi$ is the learned field estimator and $\mathcal{R}_{\text{phys}}$ denotes optional physical regularization. This formulation is natural when dense ground-truth fields are available, as in simulations, and when the sensing process provides sufficient spatial coverage to constrain the latent state.

However, in real sparse sensor networks, $\mathbf{u}(\mathbf{x}, t)$ is generally unobserved for most $\mathbf{x} \in \Omega$. The only directly observed quantities are the projections of the latent field onto the deployed sensor locations. Consequently, multiple globally distinct fields may induce identical or nearly identical sensor trajectories, making the global objective underdetermined without additional assumptions on smoothness, locality, transport, or the data-generating process. Thus, global field recovery should be interpreted as prior-guided surrogate

reconstruction rather than identifiable recovery of the true latent field under extreme sparse sensing.

**Sensor-Space Modeling.** Under extreme spatial sparsity, estimating the latent field over an arbitrary query set $\mathcal{Q} \subset \Omega \times [0, T]$ may be fundamentally underconstrained. We therefore distinguish point-estimate global field reconstruction from a more identifiable objective: *sensor-space modeling.* Let the deployed sensor network be denoted by

$$\mathcal{S} = \{\mathbf{s}_j\}_{j=1}^M, \qquad \mathbf{s}_j \in \Omega,$$

where $M$ is small relative to the spatial resolution of the underlying domain. The sensor-space observation operator is

$$\mathcal{O}_{\mathcal{S}}(\mathbf{u})(t) = \big[\mathbf{u}(\mathbf{s}_1, t), \mathbf{u}(\mathbf{s}_2, t), \ldots, \mathbf{u}(\mathbf{s}_M, t)\big] \in \mathbb{R}^{M \times q}.$$

The observed sensor trajectory is therefore

$$\mathbf{y}_t = \mathcal{O}_{\mathcal{S}}(\mathbf{u})(t) + \boldsymbol{\epsilon}_t.$$

In the longitudinal sensing regime, observations are dense in time but restricted to the spatial support of the sensor network:

$$\mathcal{D}_{\mathcal{S}} = \{(\mathbf{s}_j, t_k, \mathbf{y}_{j,k}) : j = 1, \ldots, M, \ k = 1, \ldots, T_j\}.$$

Rather than estimating $\mathbf{u}(\mathbf{x}, t)$ for arbitrary $\mathbf{x} \in \Omega$, sensor-space modeling aims to learn a predictor

$$\hat{\mathbf{u}}_{\mathcal{S}} : \mathcal{S} \times [0, T] \to \mathbb{R}^q$$

that reconstructs or forecasts the physical state only on the observational support induced by the deployed sensors:

$$\hat{\mathbf{y}}_{j,k} = \hat{\mathbf{u}}_{\mathcal{S}}(\mathbf{s}_j, t_k).$$

The corresponding learning objective is

$$\min_{\phi} \sum_{(\mathbf{s}_j, t_k, \mathbf{y}_{j,k}) \in \mathcal{D}_{\mathcal{S}}} \|f_\phi(\mathbf{s}_j, t_k; \mathcal{D}_{\mathcal{S}}) - \mathbf{y}_{j,k}\|_2^2 + \lambda \mathcal{R}_{\text{phys}}(f_\phi),$$

where $f_\phi$ is the learned reconstruction model and $\mathcal{R}_{\text{phys}}$ denotes optional physical regularization, such as PDE residuals or boundary constraints when such information is available.

Under severe sparse sensing, the most identifiable prediction target is not the unconstrained global field

$$\mathbf{u} : \Omega \times [0, T] \to \mathbb{R}^q,$$

but its restriction to the sensor support:

$$\mathbf{u}_{\mathcal{S}} = \mathbf{u}\big|_{\mathcal{S} \times [0, T]}.$$

Thus, sensor-space modeling evaluates whether a method can produce accurate, observationally grounded predictions over a persistent sensor topology, rather than claiming exact recovery of the full latent field away from observed support.

## 4 FieldFormer

### 4.1 Design Rationale

FieldFormer is designed around three core principles motivated by sparse, longitudinal spatio-temporal data:

- **Local Context with Coordinate Conditioning:** Prior work by Bhardwaj et al. (2025) has shown that purely local methods (e.g., Kriging) are effective under sparse sensing, while global neural fields can capture long-range dependencies but struggle in longitudinal regimes. FieldFormer combines local neighborhood encoding with query-coordinate and relative-offset features, enabling relational reasoning within local neighborhoods while retaining a strong local inductive bias. Although transformer attention is restricted to local neighborhoods, each prediction is conditioned on a shared continuous coordinate system through the query coordinate and token offsets, so the model can be evaluated at mesh-free locations without requiring a fixed grid or graph.

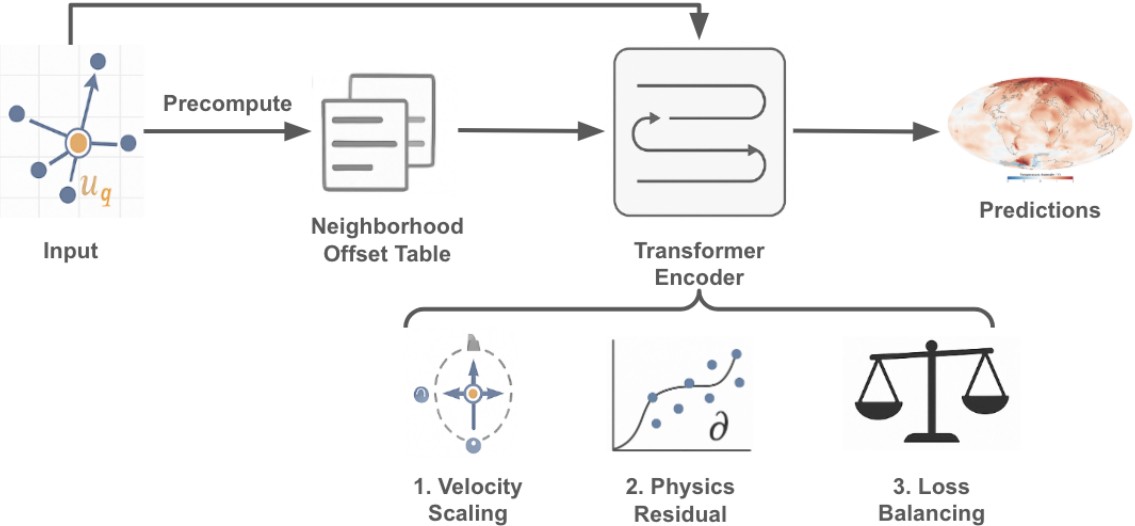

Figure 1: Conceptual overview of FieldFormer. For each query location, a physics-aware local neighborhood is constructed via learned velocity scaling, encoded with relative offsets and observations, and aggregated using a transformer encoder to produce mesh-free spatio-temporal predictions, trained with data and optionally balanced physics losses.

- **Learned Neighborhood Geometry:** Global attention incurs $O(n^2)$ memory cost, which is prohibitive for temporally dense sensing. FieldFormer instead performs attention over compact local neighborhoods, whose geometry is learned from data and physics priors. This allows the model to attend selectively to physically relevant regions while remaining memory-efficient and scalable.

- **Mesh-Free Modeling:** FieldFormer adopts a mesh-free coordinate-based representation along with a local spatio-temporal context, enabling inference at arbitrary spatial and temporal resolutions. This makes the model compatible with evolving sensor networks: when a new sensor is deployed, its measurements can be incorporated as additional coordinate-value observations without redefining the spatial grid, rebuilding a graph adjacency matrix, or changing the model architecture. Similarly, if sensors fail, move, or are added in previously unobserved regions, the model can continue operating by constructing neighborhoods from the updated set of available observations.

### 4.2 Architecture

**Local Neighborhood Encoding:** Predicting the field value at a query $(\mathbf{x}_q, t_q) \in \mathbb{R}^{d_s} \times \mathbb{R}$ requires local context: PDEs, especially with *local operators*, typically couple a state variable to its spatial and temporal neighbors rather than to the entire domain. To emulate this, FieldFormer extracts a fixed-size local neighborhood from the dataset

$$\mathcal{D} = \{(\mathbf{x}_i, t_i, \mathbf{u}_i)\}_{i=1}^N, \quad \mathbf{u}_i \in \mathbb{R}^q.$$

Neighbors are not chosen by naive Euclidean distance, but by a *velocity-scaled metric*: a learned spatio-temporal distance in which spatial and temporal offsets are rescaled before neighborhood construction:

$$d\big((\mathbf{x}_q, t_q), (\mathbf{x}_i, t_i)\big) = \sum_{k=1}^{d_s} \gamma_k^2 (x_{q,k} - x_{i,k})^2 + \gamma_t^2 (t_q - t_i)^2. \tag{2}$$

The learnable parameters $\{\gamma_k\}_{k=1}^{d_s}, \gamma_t$ weight each spatial axis and time relative to one another. They are parameterized as $\gamma = \exp(\theta)$ to enforce positivity. We call this metric velocity-scaled because these learned factors control the relative scale of spatial and temporal displacement before distance computation and

tokenization. This scaling allows the model to *discover* the anisotropy of the underlying process: for purely diffusive systems it may amplify temporal proximity, while for advective or wave-like systems it stretches neighborhoods preferentially along flow directions. Thus the model automatically learns whether a time step is "worth" more than a spatial offset.

Each neighbor $(\mathbf{x}_j, t_j, \mathbf{u}_j)$ is encoded relative to the query:

$$\mathbf{v}_j = \begin{bmatrix} \mathbf{x}_j - \mathbf{x}_q, & t_j - t_q, & \mathbf{u}_j \end{bmatrix} \in \mathbb{R}^{d_s+1+q},$$

resulting in an input matrix $\mathbf{X}_q \in \mathbb{R}^{n \times (d_s+1+q)}$. This encoding is translation-invariant and emphasizes relative differences, which is crucial when queries are at unseen coordinates.

**Sparse Neighborhood Construction.** A practical challenge in sparse longitudinal sensing is that the model must construct a local spatio-temporal context from incomplete observations without relying on a dense grid or repeatedly performing query-wise nearest-neighbor search. FieldFormer handles this by operating directly on observed sensor-time tuples. Each available observation is represented as $(\mathbf{x}_s, t_k, \mathbf{u}_{s,k})$, indexed by sensor location $s$ and time index $k$. For a query $(\mathbf{x}_q, t_q)$, FieldFormer constructs a *fixed maximal sparse context*: a fixed-size candidate set of observed sensor-time tuples formed from nearby sensors and a bounded temporal window,

$$\mathcal{C}(q) = \mathcal{S}_K(q) \times \{t_q - K_t, \ldots, t_q + K_t\},$$

where $\mathcal{S}_K(q)$ denotes a small spatial candidate set of nearby sensor locations and $K_t$ is the temporal window radius. Missing observations are excluded from this context, and the remaining valid observations are padded or truncated to produce a fixed-size set of $K$ tokens for transformer aggregation.

Here "maximal" means that the context is a candidate superset chosen before attention; attention and the prediction head still determine which included observations are useful for the query.

This fixed-context construction is a differentiable and efficient relaxation of learned metric-based neighbor selection. Given a query $q$, the learned spatio-temporal metric can be written as

$$d_q(i) = \gamma_x(q)^2 \Delta x_i^2 + \gamma_y(q)^2 \Delta y_i^2 + \gamma_t(q)^2 \Delta t_i^2,$$

where $\Delta x_i = x_i - x_q$, $\Delta y_i = y_i - y_q$, and $\Delta t_i = t_i - t_q$. Once the relevant spatial candidates are included in $\mathcal{S}_K(q)$, the temporal component of the metric is monotone in $|\Delta t_i|$. Therefore, for a sufficiently large temporal window, the fixed context contains the metric-$k$NN neighborhood as a subset:

$$\mathrm{kNN}_{d_q}(q) \subseteq \mathcal{C}(q).$$

Rather than selecting a hard metric-dependent subset whose membership changes discontinuously as $\gamma$ evolves, FieldFormer includes this local superset and lets the model learn relevance through metric-conditioned token features.

Concretely, for each candidate observation, FieldFormer computes relative offsets and scales them by positive learned factors,

$$(\widetilde{\Delta x_i}, \widetilde{\Delta y_i}, \widetilde{\Delta t_i}) = (\gamma_x \Delta x_i, \gamma_y \Delta y_i, \gamma_t \Delta t_i), \qquad \gamma = \exp(\theta),$$

with periodic wrapping for spatial coordinates when appropriate. These scaled offsets are concatenated with the observed value to form each input token. We use *token geometry* to refer to this rescaled query-relative coordinate information: each token carries both a measurement and the coordinate structure used by the local transformer. Thus, the learned geometry does not determine hard inclusion in the neighborhood; instead, it controls how the transformer interprets spatial and temporal separation within a fixed candidate context. This avoids query-wise kNN, keeps memory and computation constant per query, and provides stable gradients for the geometry parameters.

This design also reflects the information limits of sparse sensing. If the spatial candidate set fails to cover relevant spatial directions, increasing the temporal window cannot fully compensate for the missing spatial stencil information. For local PDEs, temporal samples at the same sparse locations provide useful longitudinal evidence, but they are not a substitute for unobserved spatial dependencies. FieldFormer therefore

treats the fixed context as the maximum available local evidence and uses learned geometry and attention to weight that evidence, rather than assuming that longer time histories can resolve missing spatial coverage.

**Transformer-Based Local Inference:** The neighborhood matrix $\mathbf{X}_m$ is encoded with a transformer encoder with $L$ layers:

$$\mathbf{H}_q = \text{TransformerEncoder}(\mathbf{X}_m) \in \mathbb{R}^{m \times d'},$$

where each row corresponds to a neighbor treated as a token. Because relative deltas already encode spatial and temporal positions, no global positional encoding is required. Mean pooling aggregates neighbor embeddings,

$$\mathbf{h}_q = \frac{1}{m} \sum_{j=1}^{m} \mathbf{H}_q^{(j)},$$

and a prediction head maps to the output field value:

$$\hat{\mathbf{u}}_q = \text{MLP}(\mathbf{h}_q).$$

This architecture is deliberately local: unlike global attention, its cost is constant per query, and its design aligns with the locality of PDE operators. It is also anisotropy-aware: the learned $\gamma$ scales let the same transformer backbone adjust the relative weighting of spatial and temporal offsets across regimes such as diffusion, advection, or wave propagation.

### 4.3  Inference and Efficiency

**Inference at Arbitrary Locations:** Prediction at $(\mathbf{x}, t)$ requires only three steps: 1) Use the pre-computed offset table to gather neighbors under the current scales $\gamma$. 2) Encode deltas and neighbor values into $\mathbf{X}_m$. 3) Forward through the transformer and MLP. Because neighborhoods are constant-sized, complexity per query is $\mathcal{O}(m^2 d)$, independent of total dataset size.

**Generalization and Resolution Adaptivity:** FieldFormer generalizes across resolutions: since inference is purely coordinate-based and local, the model trained at one resolution can be evaluated at another. This enables both upsampling of sparse sensor data into high-resolution fields and coarsening of predictions for downstream models.

**Scalability:** The offset-based neighbor search reduces lookup cost from $\mathcal{O}(N \log N)$ for kNN(Bentley, 1975) to amortized $\mathcal{O}(1)$. Combined with fixed-size local attention, this yields linear scaling in batch size and constant scaling with grid resolution, in stark contrast to $\mathcal{O}(N^2)$ global transformers.

### 4.4  FieldFormer Variants

**FieldFormer-PINN.  Autograd Residuals:** We treat FieldFormer itself as a *coordinate neural field*: given input $(\mathbf{x}, t)$, its prediction is differentiable w.r.t. the coordinates. Thus derivatives are obtained directly:

$$\nabla u(\mathbf{z}_q) = \frac{\partial u}{\partial \mathbf{z}}(\mathbf{z}_q), \quad \nabla^2 u(\mathbf{z}_q) = \frac{\partial^2 u}{\partial \mathbf{z}^2}(\mathbf{z}_q).$$

For governing PDE

$$\mathcal{F}(\mathbf{x}, t, u, \nabla u, \nabla^2 u, \dots) = 0,$$

the residual at a query is

$$R(\mathbf{z}_q) = \mathcal{F}\big(\mathbf{z}_q, u(\mathbf{z}_q), \nabla u(\mathbf{z}_q), \nabla^2 u(\mathbf{z}_q)\big),$$

and the physics loss is the robust Huber penalty

$$\mathcal{L}_{\text{phys}} = \frac{1}{M} \sum_{q=1}^{M} \text{Huber}(R(\mathbf{z}_q)).$$

This avoids discretization error and automatically respects the anisotropy discovered by $\gamma$.

**Loss Balancing:** To prevent one term from dominating, we normalize $\lambda_{\text{pde}}$ so that the gradient norms of data and physics terms are balanced:

$$\lambda_{\text{pde,eff}} = \lambda_{\text{pde}} \cdot \frac{\|\nabla \mathcal{L}_{\text{data}}\|}{\|\nabla \mathcal{L}_{\text{phys}}\|}.$$

Then, the final objective is

$$\mathcal{L}_{\text{total}} = \mathcal{L}_{\text{data}} + \lambda_{\text{pde,eff}} \mathcal{L}_{\text{phys}} + \lambda_{\text{bc}} \mathcal{L}_{\text{bc}}$$

where $\mathcal{L}_{\text{data}}$ is the L2 loss between the prediction and ground truth sensor observations, and $\mathcal{L}_{\text{bc}}$ is the boundary loss implemented accordingly for different boundary conditions (see details in Appendix A).

**FieldFormer–Gamma Field.** FieldFormer–Gamma Field extends the global velocity-scaled neighborhood metric by replacing fixed scale parameters with coordinate-dependent neural fields. Instead of learning a single set of global scales $\{\gamma_x, \gamma_y, \gamma_t\}$, we parameterize them as functions of the query location and time:

$$(\gamma_x(\mathbf{x}, t), \gamma_y(\mathbf{x}, t), \gamma_t(\mathbf{x}, t)) = g_\psi(\mathbf{x}, t),$$

where $g_\psi$ is a small coordinate neural network with positive outputs, for example,

$$g_\psi(\mathbf{x}, t) = \text{softplus}(\text{MLP}_\psi(\mathbf{x}, t)) + \epsilon.$$

For a query $(\mathbf{x}_q, t_q)$, neighborhoods are selected using the locally adaptive metric

$$d_q\big((\mathbf{x}_q, t_q), (\mathbf{x}_i, t_i)\big) = \gamma_x(\mathbf{x}_q, t_q)^2 (x_{q,1} - x_{i,1})^2 + \gamma_y(\mathbf{x}_q, t_q)^2 (x_{q,2} - x_{i,2})^2 + \gamma_t(\mathbf{x}_q, t_q)^2 (t_q - t_i)^2.$$

This allows FieldFormer to adapt its notion of locality across the domain: regions with faster transport, stronger diffusion, or more heterogeneous dynamics can use different spatial-temporal scales than smoother or slower regions. The resulting architecture is better suited to heterogeneous physical regimes where a single global neighborhood geometry is insufficient.

## 5 Theoretical Insights

In this section, we analyze how locality and sensor placement shape what can be recovered under sparse sensing. Our goal is to characterize when locality-aware reconstruction is well aligned with the underlying dynamics, and how missed spatial dependencies create non-identifiability under longitudinal sensing.

### 5.1 Local Structure in Spatio-Temporal Dynamics

We begin by characterizing the class of spatio-temporal processes that FieldFormer is designed to represent effectively. At a high level, a process is *local* if the state at a given location and time is primarily determined by nearby conditions in space and recent history in time, rather than by distant regions of the domain or by the entire global state. This notion of locality is a structural property of many physical systems, not an assumption introduced by FieldFormer. A formal definition in terms of finite-order differential operators is provided in Appendix D (Definition D.1).

In practice, locality appears in two complementary forms:

1. processes in which influence spreads gradually through space over time, and

2. processes in which influence propagates along paths with finite speed.

**Diffusion-like processes.** In diffusion-dominated systems, information spreads smoothly through repeated local interactions. Each location evolves based on nearby spatial context, and the effect of distant regions emerges only after many incremental updates. Examples include heat conduction or pollutant dispersion in the absence of strong advection. In such systems, accurate reconstruction depends on access to a sufficiently broad local neighborhood: missing nearby sensors can substantially degrade prediction quality, even when temporal observations are dense.

**Propagation-like processes.** In propagation-dominated systems, changes travel along preferred directions and reach downstream locations after a delay, rather than diffusing uniformly in all directions. Examples include fluid transport, traffic congestion waves, and wind-driven pollutant plumes. Here, the state at a location may be influenced by a compact set of upstream points, making anisotropic local neighborhoods effective when their geometry is captured correctly. Learning which spatial and temporal neighbors matter can therefore be more important than simply increasing neighborhood size.

In classical PDE theory, diffusion-like processes are typically modeled by parabolic equations, while propagation-dominated processes are described by hyperbolic equations. Many real-world systems combine both behaviors at different spatial and temporal scales. These locality structures explain why neighborhood-based models can reconstruct coherent sensor-space models when the sensor network sufficiently covers the relevant local domains of dependence. FieldFormer aligns with this principle by constructing adaptive local neighborhoods and learning how information should be aggregated across space and time. We next formalize this alignment through an expressivity result, followed by a miss-coverage lower bound that captures the limits imposed by sparse sensing.

## 5.2 Expressivity Under Available Local Context

We first establish that FieldFormer has sufficient representational capacity to approximate locally governed spatio-temporal dynamics when the relevant local stencil information is available. This result should be interpreted as an architectural compatibility statement. It shows that local transformer aggregation can represent finite-stencil PDE dynamics, although this setting is different from extreme sparse observations we later consider during empirical evaluation.

**Theorem 5.1** (Expressivity for Finite-Stencil Local Dynamics). *Let $u : \mathbb{R}^d \times \mathbb{R} \to \mathbb{R}^q$ solve a parabolic or hyperbolic PDE*

$$F\big(z, u(z), \nabla u(z), \dots, D^r u(z)\big) = 0, \qquad z = (x, t),$$

*with a finite-order, local operator. Assume a consistent* explicit *finite-difference scheme with compact stencil $\mathcal{S} \subset \{-s_{\max}, \dots, s_{\max}\}^d \times \{0, \dots, k\}$ and a CFL-admissible time step $\Delta t$. Let $U_h$ denote the numerical solution generated by this scheme, and write its local update as*

$$U_h^{n+1}(\mathbf{i}) = \psi(v_{\mathbf{i},n}), \qquad v_{\mathbf{i},n} = \big(U_h^{n-\ell_j}(\mathbf{i} + \mathbf{s}_j)\big)_{j=1}^N,$$

*where $\{(\mathbf{s}_j, \ell_j)\}_{j=1}^N$ is an ordering of $\mathcal{S}$ and $N = |\mathcal{S}|$. For any compact $K \subset \mathbb{R}^{d+1}$, define the compact set of stencil-value tuples*

$$\mathcal{V}_K := \big\{ v_{\mathbf{i},n} \in (\mathbb{R}^q)^N : (x_{\mathbf{i}}, t_{n+1}) \in K \big\}.$$

*Then for any $\varepsilon > 0$ there exists a FieldFormer with neighborhood size $m \geq |\mathcal{S}|$, hidden width $d'$, and depth $L$ such that*

$$\sup_{v \in \mathcal{V}_K} \|\psi(v) - \mathrm{FF}_\theta(E(v))\|_2 < \varepsilon,$$

*where $E(v)$ denotes the FieldFormer tokenization of the ordered stencil tuple $v$.*

A detailed proof of the theorem is presented in Appendix E.

*Remark* 5.2 (Implication: Alignment with Local Dynamics). Theorem 5.1 shows that FieldFormer can represent the local update structure induced by finite-stencil discretizations of parabolic and hyperbolic PDEs. It controls approximation of the discrete update map $\psi$, not convergence to the continuous PDE solution $u$. A continuous-solution statement would require an additional numerical-analysis argument, for example

$$\|u - \hat{u}\| \leq \|u - U_h\| + \|U_h - \hat{u}\|,$$

where $\|u - U_h\|$ is a discretization error controlled by assumptions such as consistency, stability, regularity, and grid refinement. The intended claim here is architectural compatibility with local finite-stencil dynamics when the relevant stencil information is available; it does not remove the ambiguity introduced when sparse sensing fails to observe important local dependencies.

### 5.3 Missed-Support Non-Identifiability under Longitudinal Sensing

Even when the underlying dynamics are representable by the model class, reconstruction accuracy is limited by what the sensor network reveals. In sparse sensing regimes, errors arise not only from model mismatch or optimization, but also from *missed spatial dependencies*: components of the local dynamics that are never observed by any sensor. This is the key distinction between expressivity and recoverability. A model may be capable of representing the correct update rule, while the observations may still be insufficient to determine which local dependencies are active.

We focus on longitudinal sensing regimes, where observations are dense in time but sparse in space. In this setting, sensors are fixed at a small subset of spatial locations and provide long temporal traces. Temporal density helps estimate local temporal evolution at the observed sites, but it does not compensate for missing spatial support. Whether a local dependency is observed or missed is therefore determined primarily by spatial sensor placement.

**Locality Assumption.** We assume that the spatio-temporal process is governed by local interactions, as defined in Appendix D (Assumption D.2). This ensures that the state update at any location depends only on a finite neighborhood in space and time.

**Sampling Assumption.** We consider a longitudinal sensing regime in which sensors are fixed at a subset of spatial locations and provide measurements across consecutive time steps. As a result, whether a local dependency is observed or missed depends on spatial sensor placement, not on temporal sampling density. We model sensor placement as selecting spatial locations uniformly at random from the set of locations influencing local dynamics. This assumption reflects common deployments such as environmental monitoring networks and is formalized in Appendix D (Assumption D.3).

**Influence Assumption.** We assume that missed spatial support can create a genuine ambiguity in the local update. To capture this, we associate an *influence weight* with each neighborhood element, with weights summing to one and reflecting relative importance. In longitudinal settings, missing a spatial location removes all of its associated temporal information, so the total influence of a spatial location is the aggregate of its temporal components. The formal set-level assumption states that, for any missed spatial set, there can be an observationally indistinguishable stencil completion whose output differs by an amount proportional to the missed influence mass. The formal statement appears in Appendix D (Assumption D.4).

Together, these assumptions formalize the central limitation of sparse sensing: even if the model class is expressive enough, unobserved spatial dependencies can leave multiple plausible local completions that agree with all measurements. Thus, failures of global reconstruction under extreme sparsity are not merely optimization failures, but consequences of the information geometry imposed by the sensor network.

**Theorem 5.3** (Missed-Support Non-Identifiability Under Longitudinal Sampling)**.** *Sample $m_x$ spatial locations uniformly without replacement from $\mathcal{S}_x$, let $I \subseteq \mathcal{S}_x$ be the sensed set and $I^c$ the missed set. Under equation 3 (in Appendix D, Assumption D.4), let $\hat{u}$ be any estimator using only the observed stencil block $V_{I \times \mathcal{T}}$. For fixed $I$, define the two completion risks*

$$\mathrm{risk}_I(V) := \mathbb{E}[\|\hat{u}(V_{I \times \mathcal{T}}) - \psi(V)\|_2 \mid I],$$

*and*

$$\mathrm{risk}_I(T_{I^c}V) := \mathbb{E}[\|\hat{u}(V_{I \times \mathcal{T}}) - \psi(T_{I^c}V)\|_2 \mid I],$$

*where $T_{I^c}V$ agrees with $V$ on the observed coordinates and differs only on the missed spatial support. Then there exists a constant $c > 0$ such that*

$$\inf_{\hat{u}} \mathbb{E}_I \left[ \max\{\mathrm{risk}_I(V), \mathrm{risk}_I(T_{I^c}V)\} \right] \ \geq \ c \, \mathbb{E}_I \left[ A(I^c) \right] \ = \ c \left( 1 - \frac{m_x}{N_x} \right).$$

A detailed proof of the theorem is presented in Appendix E.

*Remark* 5.4 (Implication: Sensor Coverage and Recoverability)**.** The theorem is a non-identifiability statement, not a universal error law for every data distribution. It says that sparse sensing can leave two plausible stencil completions that are indistinguishable from the observed data; if their updates differ by the missed

influence mass, no estimator can be accurate on both. Since longitudinal sensors provide dense time traces at fixed locations, observing a spatial site reveals all of its associated temporal information, while missing a spatial site removes all of it. The identity $\mathbb{E}_I[A(I^c)] = 1 - m_x/N_x$ captures how much influential spatial support is missed under random placement. In practice, recoverability also depends on sensor placement, noise, forcing, boundary conditions, and the inductive bias of the reconstruction model. The margin binds most naturally for transport/advective dynamics with a finite domain of dependence, and becomes vacuous when smoothness or spatial correlation makes the missed support recoverable from observed sensors (Appendix D, Remark D.5).

### 5.4 Connection to Sensor-Space Modeling

The above results clarify why sensor-space modeling is a more identifiable objective than point-estimate global field recovery under extreme sparsity. The expressivity result shows that FieldFormer is well matched to locally structured dynamics when the relevant local context is available. The missed-support non-identifiability result shows that sparse sensing can remove important spatial dependencies, leaving multiple globally plausible completions consistent with the same observations. Together, these results imply that global field recovery away from observational support is inherently prior-dependent: different models may produce different globally plausible reconstructions while remaining consistent with the same sparse sensor trajectories.

Sensor-space modeling avoids overinterpreting these globally underconstrained regions. Instead of claiming exact recovery of the full latent field, it evaluates whether a method can accurately reconstruct or forecast measurements over the persistent sensor topology. FieldFormer is designed for this setting: its adaptive local neighborhoods exploit the local domains of dependence that remain observed, while its mesh-free coordinate representation allows the same model to operate over evolving sensor networks and irregular observation layouts.

## 6 Evaluation

### 6.1 Baselines

We evaluate FieldFormer against a broad set of baselines spanning probabilistic interpolation, coordinate-based neural fields, physics-informed neural fields, and recent mesh-dependent spatio-temporal reconstruction models. This evaluation is designed to compare FieldFormer both against methods that naturally operate on continuous coordinates and against strong grid- or sensor-topology-based models that must be adapted to sparse sensing regimes.

**Mesh-free baselines.** We first compare against mesh-free methods that, like FieldFormer, can be queried at arbitrary spatio-temporal coordinates. Gaussian Processes, also referred to as Kriging in spatial statistics, provide a classical probabilistic interpolation baseline grounded in spatio-temporal kernels (Cressie, 1990). Since exact Gaussian process inference scales cubically with the number of observations, we use stochastic variational Gaussian processes (SVGPs) (Hensman et al., 2013), which approximate the posterior using inducing points while retaining flexible uncertainty-aware interpolation. In our RMSE/MAE tables, SVGP is evaluated using its posterior predictive mean, so these results measure point-prediction accuracy rather than the full quality of the predictive distribution.

We also include two coordinate-based neural field baselines. Fourier-MLPs (Tancik et al., 2020) use sinusoidal Fourier features of the input coordinates to represent high-frequency variation, while SIRENs (Sitzmann et al., 2020) use sinusoidal activations throughout the network to model oscillatory and fine-scale spatio-temporal structure. These methods learn continuous functions of the form $f(x, y, t)$ and can therefore be evaluated at arbitrary query locations, making them direct mesh-free comparators. We additionally evaluate a five-member Fourier-MLP deep ensemble trained with independent random seeds. The ensemble averages member predictions and is evaluated as a point predictor under the same RMSE/MAE protocol; although deep ensembles are often used for uncertainty-aware modeling, our tables report only ensemble-mean point accuracy.

**Physics-informed neural field baselines.** When the governing physics is available, we additionally evaluate physics-regularized versions of the neural field baselines. Specifically, we train Fourier-MLP-PINN and SIREN-PINN variants by augmenting their data-fitting objectives with PINN-style residual losses (Raissi et al., 2019). These baselines test whether global coordinate neural fields benefit from explicit physical regularization when the PDE residual can be computed. They also provide an important comparison to FieldFormer's local physics-aware formulation: while Fourier-MLP-PINN and SIREN-PINN enforce physics through global coordinate functions, FieldFormer combines local neighborhood aggregation with optional PDE-based regularization.

**Mesh-dependent spatio-temporal baselines.** Although FieldFormer is mesh-free, many recent reconstruction and imputation methods assume a fixed grid, rasterized field, or fixed sensor graph. To provide a stronger empirical comparison, we adapt three representative mesh-dependent baselines to our sparse sensing setting: RecFNO (Zhao et al., 2024), ImputeFormer (Nie et al., 2024), and Senseiver (Santos et al., 2023).

RecFNO is adapted as a grid-based field reconstruction baseline. Sparse sensor values are rasterized onto the simulation grid at the nearest grid locations, and a corresponding binary mask is provided to indicate which entries are observed. The model input consists of the sparse value grid, the observation mask, and coordinate channels, and the FNO predicts a full grid-valued field. In our implementation, this is instantiated as a Voronoi-style FNO input representation, where sparse observations and masks are placed on the grid and processed by spectral convolution layers to produce full-field predictions.

ImputeFormer is adapted as a fixed-node masked-imputation baseline over the deployed sensor network. Since the original model is designed for structured spatio-temporal imputation rather than continuous coordinate querying, we represent the persistent sensor network as a fixed set of nodes with learnable per-node embeddings and train the model over temporal windows. Because each node—including any later held out—retains its own learned embedding, ImputeFormer operates transductively over the deployed topology rather than inductively over coordinates. Inputs consist of observed values together with masks, and training randomly masks subsets of available entries so that the model learns to impute missing measurements over the fixed sensor-time array. This adaptation evaluates whether a transformer designed for fixed-node temporal imputation can recover missing sensor readings in the longitudinal sparse sensing regime.

Senseiver is adapted as a sensor-set encoder and query-decoder baseline. At each time step, available sensor measurements are encoded as a set of sensor tokens containing normalized coordinates, time, observed values, and observation masks. A latent bottleneck attends to the sensor set, and query tokens are decoded from the learned latent representation. Unlike FieldFormer, Senseiver does not construct query-specific local neighborhoods; instead, it relies on a globally regularized latent representation of the observed sensor set. This makes it a useful comparator for testing when global latent priors are beneficial relative to locality-aware reconstruction.

**Comparison scope.** These mesh-dependent baselines require adaptations that are not needed by Field-Former. RecFNO requires rasterizing sparse sensors onto a fixed grid, ImputeFormer assumes a fixed set of sensor nodes and temporal windows, and Senseiver compresses the observed sensor set into a latent representation before decoding queries. In contrast, FieldFormer operates directly on sparse coordinate-value observations and constructs adaptive local neighborhoods at query time. This mesh-free formulation gives FieldFormer several practical advantages: it supports arbitrary query coordinates, multi-resolution inference, irregular sensor layouts, and evolving sensor networks without redefining a grid, rebuilding a graph, or changing the model architecture. At the same time, including mesh-dependent baselines allows us to evaluate whether strong global or fixed-topology priors can compensate for sparse observations in regimes where global field recovery is underconstrained.

## 6.2 Experimental Setup

We evaluate all methods under sparse longitudinal sensing regimes, where observations are available at a limited number of spatial locations but over dense temporal traces. Our evaluation distinguishes between two reconstruction objectives: *sensor-space imputation* and *global field reconstruction*.

**Evaluation Regimes.** In *sensor-space imputation*, the deployed sensor topology is fixed and the goal is to predict held-out sensor-time observations from the same persistent sensor network. This setting evaluates whether a method can reconstruct missing or withheld measurements over the observational support induced by the deployed sensors. For FieldFormer, we ensure that held-out observations are not included in the local neighborhood context during training, so performance reflects genuine imputation rather than reuse of the target observation.

In *sensor-holdout evaluation*, entire sensor locations are removed from the training set and used only for validation or testing. This setting probes whether a method can generalize to unseen spatial support. Since the target locations are never observed during training, this evaluation is substantially more underconstrained than sensor-space imputation and depends strongly on the inductive priors of each model. For real-world datasets where dense full-field ground truth is unavailable, sensor-holdout evaluation serves as a practical proxy for global spatial reconstruction, since it measures prediction accuracy away from the observed training sensors.

When dense simulation fields or gridded reference data are available, we additionally evaluate *global field reconstruction* over the full spatial domain. This setting directly measures how each method extrapolates away from the observed sensor support. Because global reconstruction under extreme sparsity is generally underdetermined, we interpret these results as evaluating prior-guided surrogate reconstruction rather than identifiable recovery of the true latent field.

**Train/Validation/Test Splits.** For each dataset, we construct train, validation, and test splits in ratio $0.8 : 0.1 : 0.1$ over the sparse observations. In the standard sensor-space setting, the split is performed over observed sensor-time pairs while preserving the deployed sensor network. Thus, all sensors may appear during training, but individual observations are held out for validation and testing. In the sensor-holdout setting, validation and test sensors are removed entirely from the training set, so evaluation is performed at spatial locations unseen during training. Validation performance is used for checkpoint selection and hyperparameter tuning, while all reported numbers are computed on the held-out test split.

**Input Representation and Normalization.** All methods are trained using the same observed values, sensor masks, and data splits. For multivariate datasets, each output channel is normalized using statistics computed from the training observations only, and predictions are transformed back to the original scale before evaluation. Missing entries are excluded from both the training loss and metric computation using the corresponding observation masks. For atmospheric wind fields, we represent wind using Cartesian components rather than angular direction, avoiding discontinuities associated with circular variables.

**Training Protocol.** We use a consistent training protocol across methods wherever possible. Models are trained with AdamW, gradient clipping, validation-based checkpoint selection, and early stopping. Hyperparameters are selected using validation performance under the same data split. Mesh-free baselines operate directly on coordinate-value observations. Mesh-dependent baselines are adapted to sparse sensing through rasterization, fixed-node temporal windows, or sensor-set conditioning, as described in Section 6.1. This allows us to compare FieldFormer against both coordinate-based models and recent grid- or topology-dependent spatio-temporal reconstruction methods.

**Metrics and Uncertainty.** We report root mean squared error (RMSE) and mean absolute error (MAE) for both sensor-space prediction and global field reconstruction. Sensor-space RMSE and MAE are computed on held-out sparse sensor-time observations and serve as the primary metrics for persistent sparse sensor networks, where reliable prediction is concentrated around observational support. For datasets with dense reference fields, global RMSE and MAE are computed over the full spatial domain. For real-world datasets where dense ground-truth fields are unavailable, sensor-holdout evaluation serves as a proxy for spatial generalization beyond the observed training support.

For sensor-space metrics, we estimate uncertainty using bootstrap resampling with 1000 bootstrap samples and report mean $\pm$ standard deviation. Bootstrapping is performed over the sparse evaluation set to quantify variability in the estimated RMSE and MAE. These standard deviations measure uncertainty in the aggregate RMSE/MAE estimates under test-set resampling, not per-query predictive uncertainty, epistemic uncertainty, or uncertainty over plausible global field completions. For full-field reconstruction, we report

RMSE and MAE over the available dense reference field, using subsampling when necessary for memory efficiency.

### 6.3 Datasets

#### 6.3.1 Synthetic PDE Benchmarks

**Periodic Heat Equation.** We first evaluate on a scalar diffusion-dominated benchmark governed by the anisotropic two-dimensional heat equation with periodic boundary conditions:

$$u_t = \alpha_x u_{xx} + \alpha_y u_{yy} + f(x, y, t).$$

The forcing term $f(x, y, t)$ is a smooth sinusoidal function of space and time, inducing anisotropic and spatially coupled dynamics. The domain is discretized on a $64 \times 64$ spatial grid with 10,000 time steps, and numerical stability is enforced through the CFL condition. A smooth sinusoidal initial condition is evolved using explicit finite differences with periodic wrapping. From the resulting full field, we sample 20 sensor locations uniformly at random and extract noisy longitudinal sensor traces. This benchmark provides a controlled setting in which the governing equation, physical parameters, and boundary conditions are fully specified, allowing us to evaluate reconstruction under sparse sensing for a diffusion-like process with broad spatial coupling.

**Periodic Shallow Water Equations.** We next evaluate on a vector-valued propagation-dominated benchmark governed by the linearized two-dimensional shallow water equations with periodic boundary conditions:

$$\eta_t + H(u_x + v_y) = 0, \qquad u_t + g\eta_x = 0, \qquad v_t + g\eta_y = 0.$$

These equations describe free-surface gravity waves with characteristic speed $c = \sqrt{gH}$. We initialize the system with a Gaussian bump in surface height, which generates outward-propagating wave dynamics over the domain. The fields $(\eta, u, v)$ are simulated on a $64 \times 64$ spatial grid for 10,000 time steps using a numerically stable forward–backward scheme satisfying the CFL condition. We sample 20 sparse sensor locations and extract noisy time series from the simulated fields. This benchmark tests reconstruction of coupled multi-variable dynamics with oscillatory propagation and local wave-like dependencies.

**Advection–Diffusion Pollution Simulation.** As a synthetic-but-realistic transport benchmark, we simulate pollutant dispersion over New Delhi using a two-dimensional advection–diffusion equation:

$$u_t + \mathbf{v}(t) \cdot \nabla u = \kappa \nabla^2 u + S(x, y),$$

on a normalized $[0, 1] \times [0, 1]$ spatial domain corresponding to the city region. The initial concentration field is constructed from measurements at 32 regulatory monitoring stations on June 1, 2018, interpolated to the simulation grid using Universal Kriging. The source field combines spatial layers representing industrial activity, brick kilns, population density, and traffic intensity. Advection is driven by a synthetic monsoon-mode wind field oriented toward the northeast, with diurnal variation, directional wobble, and autoregressive stochastic gusts. Diffusion is set to produce an advection-dominated regime at the city scale, and open boundaries are handled using radiation-style boundary conditions with a sponge layer. The PDE is discretized using upwind advection, central diffusion, and a two-stage Heun/RK2 time integrator. This benchmark captures heterogeneous sources, time-varying transport, and sparse noisy sensing, while still providing dense simulated reference fields for global reconstruction evaluation.

#### 6.3.2 Real-World Sparse Sensor Benchmarks

**Real-World Pollution Monitoring.** We evaluate on a real-world air-quality monitoring dataset from New Delhi. The dataset uses the city's deployed monitoring network, consisting of 32 air-quality stations distributed over an area of approximately $858 \text{ km}^2$. These stations are operated by public monitoring agencies, including the Central Pollution Control Board (CPCB), Delhi Pollution Control Committee (DPCC), and the Indian Meteorological Department (IMD). We use hourly measurements collected over a 30-month period from May 1, 2018 to November 1, 2020. The prediction targets are $PM_{2.5}$ and $PM_{10}$ concentrations.

This dataset represents a persistent sparse sensor network: measurements are temporally dense, but spatial coverage is limited to a small number of fixed monitoring locations. Since dense ground-truth concentration fields are unavailable, we evaluate sensor-space imputation and sensor-holdout generalization.

**Real-World Atmospheric Measurements.** We also evaluate on a real-world atmospheric sensing dataset using the same New Delhi sensor locations and time period as the pollution monitoring dataset. Instead of pollutant concentrations, this dataset contains meteorological variables measured over the deployed monitoring network. We model four atmospheric variables: average temperature, relative humidity, and horizontal wind components derived from wind speed and direction. Representing wind as Cartesian components avoids discontinuities associated with angular direction. This benchmark tests multivariate sparse sensor modeling for coupled atmospheric fields, where variables exhibit different smoothness, transport, and temporal variation patterns. As with the real-world pollution dataset, dense full-field ground truth is unavailable, so we evaluate sensor-space imputation and use sensor-holdout evaluation as a proxy for spatial generalization beyond the observed training sensors.

## 6.4 Results

### 6.4.1 Sensor-Space Imputation

We first evaluate sensor-space imputation, where the goal is to reconstruct held-out sensor-time observations over a persistent sensing topology. This is the most observationally grounded evaluation regime under extreme sparsity, since predictions are restricted to locations supported by the deployed sensor network.

Table 1 reports sensor-space performance on the three synthetic PDE benchmarks. FieldFormer is consistently competitive across all three systems, achieving near-best performance on Heat, Pollution, and SWE. On Heat and Pollution, Fourier-MLP obtains slightly lower sparse RMSE/MAE, while FieldFormer remains close. On SWE, FieldFormer is nearly tied with SIREN on sparse sensor-space metrics while substantially outperforming several mesh-dependent and globally regularized baselines. These results indicate that locality-aware reconstruction is well suited to sparse sensor imputation across diffusion, wave, and transport regimes.

Table 1: Sensor-space imputation results on synthetic PDE benchmarks. Results are reported as mean $\pm$ std over bootstrap samples. Lower is better.

| Model | Heat RMSE | Heat MAE | Pollution RMSE | Pollution MAE | SWE RMSE | SWE MAE |
|---|---|---|---|---|---|---|
| FieldFormer | 0.09888 ± 0.0007416 | 0.07876 ± 0.0005815 | 0.1154 ± 0.00032 | 0.09216 ± 0.0002176 | 0.04644 ± 0.000254 | 0.03703 ± 0.0001957 |
| Fourier-MLP | **0.09786 ± 0.0006985** | **0.07786 ± 0.0005517** | **0.1147 ± 0.0003252** | **0.09153 ± 0.0002235** | 0.1011 ± 0.0004408 | 0.08049 ± 0.0002947 |
| Fourier-MLP Ensemble | 0.09807 ± 0.0006568 | 0.07799 ± 0.0005219 | 0.1289 ± 0.0005315 | 0.1004 ± 0.0002781 | 0.1010 ± 0.0004365 | 0.08043 ± 0.0002840 |
| Fourier-MLP-PINN | 0.1584 ± 0.001396 | 0.1206 ± 0.0008432 | 0.1148 ± 0.0003401 | 0.09159 ± 0.0002141 | 0.101 ± 0.0004342 | 0.08044 ± 0.0002824 |
| SIREN | 0.09883 ± 0.0006853 | 0.07877 ± 0.0004833 | 0.1162 ± 0.0003095 | 0.0928 ± 0.0002089 | **0.04623 ± 0.0002687** | **0.03683 ± 0.0001931** |
| SIREN-PINN | 0.1793 ± 0.0009729 | 0.1434 ± 0.0007186 | 0.1158 ± 0.0002912 | 0.09245 ± 0.000185 | 0.1008 ± 0.0004355 | 0.08029 ± 0.000281 |
| SVGP | 0.1185 ± 0.0008308 | 0.09252 ± 0.0006093 | 0.1171 ± 0.0003127 | 0.0934 ± 0.0003083 | 0.101 ± 0.0004366 | 0.08044 ± 0.0002813 |
| RecFNO | 0.1038 ± 0.0006049 | 0.08269 ± 0.0004322 | 0.1173 ± 0.0002567 | 0.09375 ± 0.0002272 | 0.06296 ± 0.0005098 | 0.04939 ± 0.0004467 |
| ImputeFormer | 0.1058 ± 0.0006532 | 0.08423 ± 0.000568 | 0.1185 ± 0.0003339 | 0.09444 ± 0.0002698 | 0.06512 ± 0.0005653 | 0.05111 ± 0.0004587 |
| Senseiver | 0.1011 ± 0.0007449 | 0.08045 ± 0.0005726 | 0.1163 ± 0.0002625 | 0.09294 ± 0.0001972 | 0.07722 ± 0.0005243 | 0.06056 ± 0.0003741 |

Table 2 reports sensor-space imputation results on the real-world atmospheric and pollution monitoring datasets. FieldFormer performs especially strongly in this regime. On atmospheric measurements, Field-Former achieves the best or near-best performance across temperature, relative humidity, and wind components, with particularly large gains over Fourier-MLP, the Fourier-MLP ensemble, SIREN, SVGP, ImputeFormer, and Senseiver on temperature and relative humidity. The ensemble modestly improves over single Fourier-MLP on most real-world sensor-space variables, but remains well behind FieldFormer. On the real-world pollution dataset, FieldFormer obtains the lowest RMSE and MAE for both $PM_{10}$ and $PM_{2.5}$, showing that locality-aware modeling is effective for persistent urban monitoring networks.

Overall, the sensor-space results support the main intended use case of FieldFormer: imputation over persistent sparse sensor networks. FieldFormer is not uniformly best on every synthetic sensor-space metric, but it is consistently competitive and performs particularly strongly on real-world sparse sensing datasets, where fixed sensor topology and local spatial structure are central.

Table 2: Sensor-space imputation results on real-world sparse sensor benchmarks. Results are reported as mean ± std over bootstrap samples. Lower is better.

| Dataset | Model | Variable 1 RMSE | Variable 1 MAE | Variable 2 RMSE | Variable 2 MAE |
|---------|-------|-----------------|----------------|-----------------|----------------|
| Atmospheric: AT/RH | FieldFormer | **0.9966 ± 0.02692** | **0.5198 ± 0.004008** | **4.333 ± 0.0544** | **2.171 ± 0.01099** |
| | Fourier-MLP | 3.899 ± 0.01267 | 3.157 ± 0.009728 | 14.94 ± 0.03613 | 11.88 ± 0.03015 |
| | Fourier-MLP Ensemble | 3.878 ± 0.01237 | 3.144 ± 0.009327 | 14.87 ± 0.03755 | 11.83 ± 0.03070 |
| | SIREN | 4.125 ± 0.01249 | 3.333 ± 0.009027 | 16.28 ± 0.03885 | 13.03 ± 0.03152 |
| | SVGP | 8.494 ± 0.02936 | 6.905 ± 0.02905 | 23.47 ± 0.06189 | 19.42 ± 0.05223 |
| | RecFNO | 1.449 ± 0.02407 | 0.9899 ± 0.005657 | 5.992 ± 0.05269 | 3.694 ± 0.015 |
| | ImputeFormer | 1.619 ± 0.02632 | 1.056 ± 0.005216 | 6.978 ± 0.06032 | 4.346 ± 0.02544 |
| | Senseiver | 2.243 ± 0.02496 | 1.557 ± 0.009949 | 9.56 ± 0.08187 | 6.341 ± 0.02758 |
| Atmospheric: $U_x/V_y$ | FieldFormer | 0.6807 ± 0.0134 | **0.3213 ± 0.002875** | **0.7011 ± 0.01045** | **0.3468 ± 0.002512** |
| | Fourier-MLP | 0.8519 ± 0.008105 | 0.5463 ± 0.002747 | 0.8375 ± 0.00963 | 0.5132 ± 0.002926 |
| | Fourier-MLP Ensemble | 0.8412 ± 0.007255 | 0.5357 ± 0.002651 | 0.8290 ± 0.009320 | 0.5049 ± 0.002972 |
| | SIREN | 1.041 ± 0.005132 | 0.702 ± 0.002925 | 0.959 ± 0.007708 | 0.5865 ± 0.00253 |
| | SVGP | 1.357 ± 0.01282 | 0.8861 ± 0.004546 | 1.086 ± 0.009182 | 0.6687 ± 0.00327 |
| | RecFNO | **0.6653 ± 0.01107** | 0.3558 ± 0.002152 | 0.7204 ± 0.007887 | 0.3801 ± 0.001811 |
| | ImputeFormer | 0.7271 ± 0.01181 | 0.388 ± 0.002308 | 0.7361 ± 0.01278 | 0.399 ± 0.002718 |
| | Senseiver | 0.8421 ± 0.01156 | 0.5182 ± 0.00315 | 0.8082 ± 0.009127 | 0.5003 ± 0.002084 |
| Pollution: $PM_{10}/PM_{2.5}$ | FieldFormer | **39.25 ± 0.8496** | **20.28 ± 0.1172** | **22.06 ± 0.664** | **11.23 ± 0.0691** |
| | Fourier-MLP | 83.48 ± 0.3978 | 54.81 ± 0.1662 | 53.19 ± 0.4248 | 31.74 ± 0.1232 |
| | Fourier-MLP Ensemble | 81.40 ± 0.3946 | 53.25 ± 0.1711 | 51.80 ± 0.4218 | 30.75 ± 0.1193 |
| | SIREN | 92.81 ± 0.3154 | 62.41 ± 0.1189 | 58.95 ± 0.4278 | 35.93 ± 0.1205 |
| | SVGP | 147.9 ± 0.5388 | 110 ± 0.3128 | 98.19 ± 0.4255 | 67.41 ± 0.2203 |
| | RecFNO | 53.84 ± 0.7037 | 31.25 ± 0.19 | 32.91 ± 0.7806 | 17.65 ± 0.1067 |
| | ImputeFormer | 58.96 ± 0.6289 | 35.2 ± 0.1462 | 36.27 ± 0.7988 | 19.09 ± 0.1309 |
| | Senseiver | 60.05 ± 0.5948 | 36.94 ± 0.1758 | 36.21 ± 0.6968 | 19.98 ± 0.08917 |

### 6.4.2 Global Field Reconstruction

We next evaluate global field reconstruction, where the goal is to predict values away from the observed sensor support. For synthetic datasets, dense reference fields are available, allowing direct full-field evaluation. Table 3 merges the full-field results for Heat, Pollution, and SWE.

The results show that global reconstruction is substantially more variable than sensor-space imputation. No single method dominates across all datasets and metrics. FieldFormer remains competitive across all three benchmarks, but different baselines are favored in different regimes: RecFNO performs well on Heat full-field reconstruction, Fourier-MLP and SVGP are competitive on Pollution, and Fourier-MLP-PINN/SVGP-style smooth reconstructions perform strongly on SWE full-field metrics. This variability supports our central argument that global reconstruction under extreme sparse sensing is prior-dependent: away from observed sensors, performance depends strongly on how each model's inductive bias aligns with the underlying data-generating process.

Table 3: Global field reconstruction results on synthetic PDE benchmarks. Dense reference fields are available for these datasets. Lower is better.

| Model | Heat RMSE | Heat MAE | Pollution RMSE | Pollution MAE | SWE RMSE | SWE MAE |
|-------|-----------|----------|----------------|---------------|----------|---------|
| FieldFormer | 0.1818 ± 0.001179 | 0.1313 ± 0.0008449 | 0.4359 ± 0.04525 | 0.0831 ± 0.006115 | 0.2015 ± 0.0002546 | 0.1465 ± 0.0002115 |
| Fourier-MLP | 0.2146 ± 0.001386 | 0.1469 ± 0.0009123 | **0.4326 ± 0.04487** | 0.09857 ± 0.005979 | 0.1807 ± 0.0001687 | 0.1319 ± 0.0001417 |
| Fourier-MLP Ensemble | 0.2089 ± 0.001412 | 0.1414 ± 0.0009312 | 0.4344 ± 0.04490 | 0.09691 ± 0.006017 | 0.1828 ± 0.0001669 | 0.1340 ± 0.0001324 |
| Fourier-MLP-PINN | 0.2141 ± 0.001346 | 0.1517 ± 0.001026 | 0.4652 ± 0.04322 | 0.1314 ± 0.006371 | 0.18 ± 0.0001695 | 0.1311 ± 0.0001448 |
| SIREN | 0.1783 ± 0.0009598 | 0.1328 ± 0.0007279 | 0.4556 ± 0.04306 | 0.1256 ± 0.006149 | 0.6818 ± 0.000437 | 0.473 ± 0.0003094 |
| SIREN-PINN | 0.195 ± 0.0009061 | 0.1533 ± 0.0007207 | 0.4598 ± 0.0429 | 0.122 ± 0.006211 | **0.1799 ± 0.0001696** | **0.131 ± 0.0001448** |
| SVGP | 0.1788 ± 0.001214 | 0.1306 ± 0.0008949 | 0.4349 ± 0.04505 | **0.08208 ± 0.006107** | 0.18 ± 0.0001695 | 0.1311 ± 0.0001449 |
| RecFNO | **0.1771 ± 0.001173** | **0.1212 ± 0.0007414** | 0.5676 ± 0.03491 | 0.1684 ± 0.006464 | 0.1985 ± 0.0002363 | 0.15 ± 0.0001697 |
| ImputeFormer | 0.1916 ± 0.001001 | 0.1435 ± 0.000781 | 0.4482 ± 0.0439 | 0.1103 ± 0.00613 | 0.1816 ± 0.0001737 | 0.1321 ± 0.0001524 |
| Senseiver | 0.1959 ± 0.002121 | 0.1421 ± 0.001958 | 0.4429 ± 0.00875 | 0.09243 ± 0.001077 | 0.1867 ± 0.0006654 | 0.1364 ± 0.0004551 |

For real-world datasets, dense full-field ground truth is unavailable. We therefore use sensor-holdout evaluation as a proxy for spatial generalization: entire sensors are removed from training and used only for testing. Table 4 reports these results for atmospheric and pollution monitoring datasets. This setting should be interpreted with some care because the baselines expose different inductive biases and query interfaces. In particular, ImputeFormer is a fixed-node imputer over the deployed sensor array, giving it access to a transductive fixed-topology structure (a dedicated learned embedding per held-out node, with only the held-out

values withheld) that is advantageous when the held-out targets are drawn from the same persistent sensor network.

The real-world sensor-holdout results further demonstrate the prior-dependent nature of spatial extrapolation. ImputeFormer performs strongly in this setting, especially on relative humidity, wind-$U_x$, and both pollution variables, but this should be read together with its fixed-node/transductive advantage. Senseiver is also competitive, obtaining the best atmospheric temperature RMSE, but it is not consistently best across variables or metrics. Ensembling the Fourier-MLP helps on some sensor-holdout metrics but does not produce a uniformly strong extrapolator. FieldFormer is less dominant in this setting than in sensor-space imputation, consistent with the fact that sensor-holdout evaluation removes the local observational support that locality-aware methods rely on. Overall, Table 4 reinforces the same conclusion as the synthetic full-field results: no model uniformly solves reconstruction away from observed support; performance depends on how each model's prior matches the dataset and evaluation regime.

Table 4: Sensor-holdout results on real-world sparse sensor benchmarks, used as a proxy for spatial generalization when dense full-field ground truth is unavailable. Results are reported as mean ± std over bootstrap samples. Lower is better. ImputeFormer is instantiated over the full deployed node set and holds a learnable per-node embedding for the held-out validation/test sensors; only their observed values are withheld (masked in the input and excluded from the loss). It therefore imputes at known nodes of a fixed topology using node-specific learned parameters, rather than predicting at unseen coordinates as the coordinate-query baselines do, so its results here are not strictly comparable to the coordinate-query methods.

| Dataset | Model | Variable 1 RMSE | Variable 1 MAE | Variable 2 RMSE | Variable 2 MAE |
|---|---|---|---|---|---|
| Atmospheric: AT/RH | FieldFormer | 4.63 ± 0.6376 | 3.14 ± 0.4384 | 15.75 ± 2.517 | 10.04 ± 1.568 |
| | Fourier-MLP | 5.403 ± 0.3827 | 4.367 ± 0.2752 | 27.33 ± 2.096 | 22.55 ± 1.938 |
| | Fourier-MLP Ensemble | 5.091 ± 0.3700 | 4.039 ± 0.2682 | 18.68 ± 0.9145 | 15.31 ± 0.8338 |
| | SIREN | 6.353 ± 0.4423 | 5.045 ± 0.3822 | 21.16 ± 1.12 | 17.27 ± 1.018 |
| | SVGP | 8.795 ± 0.5025 | 7.144 ± 0.4788 | 22.01 ± 0.9019 | 18.54 ± 0.8679 |
| | RecFNO | 11.34 ± 0.7088 | 9.333 ± 0.59 | 21.88 ± 1.137 | 18.2 ± 1.039 |
| | ImputeFormer | 3.037 ± 0.6049 | **2.001 ± 0.3392** | **7.208 ± 0.611** | **5.327 ± 0.3565** |
| | Senseiver | **3.017 ± 0.5426** | 2.042 ± 0.3465 | 10.59 ± 1.065 | 8.112 ± 0.8772 |
| Atmospheric: $U_x/V_y$ | FieldFormer | 2.26 ± 0.6081 | 1.282 ± 0.2269 | 1.502 ± 0.1426 | 1.026 ± 0.09529 |
| | Fourier-MLP | 2.162 ± 0.7587 | 1.183 ± 0.2525 | 0.9047 ± 0.1635 | 0.5812 ± 0.0651 |
| | SIREN | 2.101 ± 0.7676 | 1.027 ± 0.2498 | 0.9368 ± 0.1635 | 0.614 ± 0.0696 |
| | Fourier-MLP Ensemble | 2.015 ± 0.7490 | 0.9600 ± 0.2368 | **0.8970 ± 0.1819** | 0.5455 ± 0.07301 |
| | SVGP | 2.033 ± 0.7474 | 0.9731 ± 0.2393 | 0.9001 ± 0.1931 | **0.5331 ± 0.07598** |
| | RecFNO | 2.034 ± 0.7 | 0.9904 ± 0.2369 | 1.014 ± 0.1774 | 0.6318 ± 0.07666 |
| | ImputeFormer | **1.911 ± 0.8183** | **0.7754 ± 0.2425** | 0.9473 ± 0.1837 | 0.5974 ± 0.08248 |
| | Senseiver | 2.091 ± 0.687 | 1.001 ± 0.2517 | 1.447 ± 0.2026 | 0.9016 ± 0.1555 |
| Pollution: $PM_{10}/PM_{2.5}$ | FieldFormer | 115 ± 13.11 | 69.63 ± 9.17 | 45.95 ± 4.77 | 26.76 ± 3.152 |
| | Fourier-MLP | 121.7 ± 9.278 | 82.31 ± 6.506 | 67.77 ± 6.211 | 42.41 ± 4.393 |
| | Fourier-MLP Ensemble | 120.8 ± 8.308 | 82.58 ± 6.140 | 68.39 ± 5.787 | 44.01 ± 4.061 |
| | SIREN | 149.4 ± 12.78 | 103.6 ± 8.571 | 82.73 ± 7.859 | 54.26 ± 4.981 |
| | SVGP | 166.5 ± 12.13 | 122 ± 7.467 | 99.17 ± 9.501 | 68.2 ± 6.101 |
| | RecFNO | 170.9 ± 15.06 | 127.4 ± 11.63 | 75.75 ± 5.656 | 52.74 ± 2.747 |
| | ImputeFormer | **80.76 ± 10.07** | **47.09 ± 6.202** | **38.93 ± 4.384** | **21.15 ± 2.274** |
| | Senseiver | 86.51 ± 11.83 | 50.6 ± 7.264 | 39.84 ± 4.632 | 22.77 ± 2.687 |

Taken together, the global reconstruction and sensor-holdout results show that extrapolating away from observed sensor support is substantially more sensitive to the choice of inductive prior than imputing over an existing sensor network. FieldFormer performs strongly when local sensor support is available, fixed-node imputers can benefit from transductive topology in sensor-holdout settings, and globally regularized baselines can be advantageous when their priors align with the dataset. The absence of a uniform winner supports our broader framing: under extreme sparse sensing, global reconstruction should be treated as prior-guided surrogate completion, whereas sensor-space imputation is the more identifiable and practically reliable objective.

**Inference efficiency.** Appendix C reports wall-clock sparse-test prediction time per query, peak GPU memory, and setup time using the same trained checkpoints and sparse-test protocol. The efficiency comparison is most meaningful against context-conditioned reconstruction models, since direct coordinate fields

such as Fourier-MLP and SIREN do not aggregate observed sensor context and therefore do not incur attention or set-conditioning costs. In this comparable regime, FieldFormer is an efficient transformer architecture: its fixed-size local attention is consistently over 100× faster per query than Senseiver's global sensor-set transformer and roughly 7–10× faster than the grid-based RecFNO evaluator across the timing benchmarks, while retaining mesh-free coordinate querying. ImputeFormer has lower query-time latency after setup, but this is because its fixed-node/transductive formulation precomputes and caches predictions over the deployed sensor-time array, reducing evaluation to lookup rather than arbitrary coordinate reconstruction. These results support FieldFormer's design goal: efficient transformer-based sparse reconstruction through local context rather than global attention over the observation set.

## 6.5 Ablation Studies

We ablate three FieldFormer variants to isolate the role of physics regularization, transformer-based local aggregation, and spatially adaptive neighborhood geometry. **FieldFormer-PINN** augments FieldFormer with the PDE residual and boundary losses described in Section 4.4. **FieldFormer-Gamma Field** replaces global velocity-scaling parameters with coordinate-dependent gamma fields, also described in Section 4.4. **FieldFormer-MLP** preserves the same mesh-free neighborhood construction and velocity-scaled spatio-temporal metric as FieldFormer, but replaces the local transformer encoder with a multilayer perceptron. In this variant, the selected local neighborhood is encoded through relative offsets and observed values, then aggregated by an MLP prediction head rather than attention. This ablation tests whether performance gains arise primarily from adaptive local evidence selection or from transformer-based relational reasoning within the neighborhood.

These ablations should be interpreted together with Senseiver from Section 6.1. The comparison between FieldFormer and FieldFormer-MLP isolates the effect of the local transformer: both models use the same query-specific local decomposition, the same sparse sensor-time neighborhood, and the same relative-coordinate tokenization, but FieldFormer-MLP replaces attention with an MLP aggregator. Senseiver provides the complementary no-local-decomposition attention comparator. It is still a transformer-based sensor-set model, but it conditions on the observed sensor set globally rather than constructing a query-specific local neighborhood before prediction. Thus, the three-way comparison FieldFormer vs. FieldFormer-MLP vs. Senseiver separates two design choices: whether the model first decomposes reconstruction into local query neighborhoods, and whether the evidence within that context is aggregated by attention or by an MLP. The real-world FieldFormer-MLP results in Table 7, together with the Senseiver results in Tables 2 and 4, show that both choices matter: local neighborhood construction alone is competitive, but FieldFormer's local attention improves robustness, while global attention without the local decomposition is not uniformly stronger under sparse sensing.

**Synthetic PDE benchmarks.** Table 5 reports sensor-space imputation results on the three synthetic PDE benchmarks. FieldFormer without explicit physics regularization performs best on Heat and SWE sensor-space prediction, while FieldFormer-Gamma Field is marginally strongest on the Pollution simulation. FieldFormer-MLP is competitive on Heat but degrades more noticeably on Pollution and SWE, suggesting that attention-based aggregation is useful when local neighborhoods contain coupled or directional structure. FieldFormer-PINN generally worsens sensor-space performance, especially on Heat and SWE, indicating that physics residuals can conflict with sparse noisy observations when the reconstruction target is sensor-space imputation.

Table 5: Ablation study on synthetic PDE benchmarks: sensor-space imputation. Results are reported as mean ± std over bootstrap samples. Lower is better.

| Model | Heat RMSE | Heat MAE | Pollution RMSE | Pollution MAE | SWE RMSE | SWE MAE |
|---|---|---|---|---|---|---|
| FieldFormer-PINN | $0.1071 \pm 0.0009743$ | $0.08513 \pm 0.000793$ | $0.1214 \pm 0.0003221$ | $0.09683 \pm 0.0002315$ | $0.05716 \pm 0.0002378$ | $0.04541 \pm 0.0002047$ |
| FieldFormer | $\mathbf{0.09888 \pm 0.0007416}$ | $\mathbf{0.07876 \pm 0.0005815}$ | $\mathbf{0.1154 \pm 0.00032}$ | $0.09216 \pm 0.0002176$ | $\mathbf{0.04644 \pm 0.000254}$ | $\mathbf{0.03703 \pm 0.0001957}$ |
| FieldFormer-Gamma Field | $0.225 \pm 0.001009$ | $0.1741 \pm 0.0006781$ | $\mathbf{0.1154 \pm 0.0002949}$ | $\mathbf{0.09212 \pm 0.0001884}$ | $0.04696 \pm 0.0002675$ | $0.03738 \pm 0.000196$ |
| FieldFormer-MLP | $0.09908 \pm 0.000686$ | $0.07895 \pm 0.0005192$ | $0.1157 \pm 0.0003294$ | $0.09245 \pm 0.0002116$ | $0.04713 \pm 0.0002381$ | $0.03761 \pm 0.000187$ |

Table 6 reports global field reconstruction results on the synthetic datasets. Unlike sensor-space imputation, the best variant changes across datasets. FieldFormer-MLP performs best on Heat, FieldFormer and

FieldFormer-Gamma Field are strongest on different Pollution metrics, and FieldFormer-Gamma Field gives the best global reconstruction on SWE among FieldFormer variants. This supports the broader observation that global reconstruction is more prior-dependent than sensor-space imputation: architectural changes that help away from sensors do not necessarily improve sensor-space prediction.

Table 6: Ablation study on synthetic PDE benchmarks: global field reconstruction. Dense reference fields are available for these datasets. Lower is better.

| Model | Heat RMSE | Heat MAE | Pollution RMSE | Pollution MAE | SWE RMSE | SWE MAE |
|---|---|---|---|---|---|---|
| FieldFormer-PINN | $0.1641 \pm 0.001102$ | $0.1189 \pm 0.0007892$ | $0.4582 \pm 0.04409$ | $0.1386 \pm 0.00625$ | $0.2374 \pm 0.00019$ | $0.1717 \pm 0.000157$ |
| FieldFormer | $0.1818 \pm 0.001179$ | $0.1313 \pm 0.0008449$ | $\mathbf{0.4359 \pm 0.04525}$ | $0.0831 \pm 0.006115$ | $0.2015 \pm 0.0002546$ | $0.1465 \pm 0.0002115$ |
| FieldFormer-Gamma Field | $0.2193 \pm 0.001191$ | $0.1641 \pm 0.0008903$ | $0.4366 \pm 0.04515$ | $\mathbf{0.08203 \pm 0.006125}$ | $\mathbf{0.1874 \pm 0.0001986}$ | $\mathbf{0.1362 \pm 0.0001916}$ |
| FieldFormer-MLP | $\mathbf{0.1594 \pm 0.0009926}$ | $\mathbf{0.1144 \pm 0.0007596}$ | $0.4645 \pm 0.04324$ | $0.1455 \pm 0.006278$ | $0.219 \pm 0.000232$ | $0.1577 \pm 0.0001574$ |

**Real-world sparse sensor benchmarks.** For real-world datasets, we compare FieldFormer against FieldFormer-Gamma Field and FieldFormer-MLP, while excluding PINN variants because no reliable real-world physics residual is available. Table 7 shows sensor-space imputation results on atmospheric and pollution monitoring datasets. FieldFormer-MLP tests whether sparse local neighborhood construction alone is sufficient when the local transformer is replaced by an MLP aggregator. Gamma Field improves several wind and pollution metrics, but does not uniformly improve all atmospheric variables. In particular, FieldFormer remains better for relative humidity and some mean absolute error metrics. This suggests that coordinate-dependent neighborhood geometry can help when local scales vary spatially, but also introduces additional flexibility that may not always be beneficial under sparse real-world sensing.

Table 7: Ablation study on real-world benchmarks: sensor-space imputation. Results are reported as mean ± std over bootstrap samples. Lower is better.

| Dataset | Variable | FieldFormer RMSE | FieldFormer MAE | Gamma Field RMSE | Gamma Field MAE | FieldFormer-MLP RMSE | FieldFormer-MLP MAE |
|---|---|---|---|---|---|---|---|
| Atmospheric | AT | $0.9966 \pm 0.02692$ | $\mathbf{0.5198 \pm 0.004008}$ | $\mathbf{0.9935 \pm 0.02355}$ | $0.5232 \pm 0.004039$ | $1.025 \pm 0.02321$ | $0.5712 \pm 0.004604$ |
| Atmospheric | RH | $\mathbf{4.333 \pm 0.0544}$ | $\mathbf{2.171 \pm 0.01099}$ | $4.498 \pm 0.07141$ | $2.18 \pm 0.01716$ | $4.425 \pm 0.04953$ | $2.267 \pm 0.01041$ |
| Atmospheric | $U_x$ | $0.6807 \pm 0.0134$ | $0.3213 \pm 0.002875$ | $\mathbf{0.6547 \pm 0.009111}$ | $\mathbf{0.3189 \pm 0.002666}$ | $0.6888 \pm 0.01342$ | $0.3436 \pm 0.003317$ |
| Atmospheric | $V_y$ | $0.7011 \pm 0.01045$ | $0.3468 \pm 0.002512$ | $\mathbf{0.6997 \pm 0.009647}$ | $\mathbf{0.3457 \pm 0.002271}$ | $0.7388 \pm 0.009629$ | $0.3717 \pm 0.00248$ |
| Pollution | $PM_{10}$ | $39.25 \pm 0.8496$ | $20.28 \pm 0.1172$ | $\mathbf{38.83 \pm 0.9117}$ | $\mathbf{19.91 \pm 0.1438}$ | $40.23 \pm 0.7156$ | $21.23 \pm 0.1219$ |
| Pollution | $PM_{2.5}$ | $22.06 \pm 0.664$ | $11.23 \pm 0.0691$ | $\mathbf{21.99 \pm 0.72}$ | $\mathbf{11.01 \pm 0.07286}$ | $23.24 \pm 0.6644$ | $11.65 \pm 0.06181$ |

Overall, the ablations show that FieldFormer's core locality-aware architecture is already strong for sensor-space imputation. Physics regularization does not consistently improve sparse prediction and can degrade performance when the residual objective conflicts with noisy or underconstrained observations. Replacing the transformer with an MLP preserves some benefits of adaptive neighborhood construction and remains competitive on several real-world variables, but it underperforms the transformer-based variants on most metrics. This suggests that attention-based relational aggregation improves reliability in heterogeneous real-world sensor data. Gamma Field improves selected global and heterogeneous real-world metrics, especially for some pollution and wind variables, but its gains are not uniform, suggesting that coordinate-dependent geometry is useful but should be evaluated on a case-by-case basis.

# 7 Discussion and Conclusion

This work studies sparse spatio-temporal reconstruction under the limits imposed by persistent sensor networks. Rather than treating sparse observations as sufficient for exact global field recovery, we argue that extreme spatial sparsity induces an observability gap: many globally distinct latent fields may remain consistent with the same sensor-space measurements. In this setting, reconstruction quality depends not only on model capacity, but also on the alignment between the model's inductive bias and the information geometry induced by the sensor network.

FieldFormer is designed around this perspective. By operating directly on coordinate-value observations, it avoids dependence on fixed grids or static graphs and supports mesh-free inference over irregular sensing layouts. Its metric-scaled local neighborhoods and local transformer aggregation align reconstruction with localized domains of dependence, making the model particularly effective for sensor-space imputation over persistent sparse sensor networks. Across synthetic PDE benchmarks and real-world monitoring datasets,

FieldFormer performs strongly in this regime, especially when the target is to reconstruct missing or withheld measurements over an existing sensing topology.

Our results also clarify the distinction between sensor-space imputation and global field reconstruction. Sensor-space imputation is comparatively well grounded because predictions are made on the observational support of the deployed sensors. In contrast, global reconstruction away from observed support is substantially more prior-dependent. Across datasets, different methods perform best in different global reconstruction or sensor-holdout settings, with no universally dominant inductive bias. This variability supports the central claim of the paper: under extreme sparse sensing, global field reconstruction should be interpreted as prior-guided surrogate completion rather than identifiable recovery of the true latent physical state.

The ablation studies further highlight the role of FieldFormer's architectural choices. Replacing the local transformer with an MLP preserves the benefit of adaptive neighborhood construction but is less reliable on coupled or transport-dominated systems, suggesting that relational aggregation within local neighborhoods is important. Adding PINN-style physics regularization does not consistently improve performance and can degrade sensor-space accuracy, indicating that physics losses may conflict with sparse noisy observations when the residual objective is weakly constrained. Coordinate-dependent Gamma Fields improve selected metrics in heterogeneous regimes, but their gains are not uniform, suggesting that local geometry adaptation is a useful extension rather than a universally better default.

The learned velocity-scaled metric used by FieldFormer should also be distinguished from characteristic-aligned transport coordinates such as $\Delta x - v\Delta t$. When a reliable velocity field $v$ is known, characteristic coordinates can be a useful specialization for wave-like or advective systems. Our setting is broader: in diffusion-dominated systems, mixed advection–diffusion processes, and real-world monitoring datasets, a single known transport velocity may be unavailable, spatially heterogeneous, variable-dependent, or not well defined. FieldFormer therefore learns anisotropic spatial and temporal scales without assuming a prescribed relation $\Delta x = v\Delta t$. Incorporating characteristic-aligned metrics when reliable velocity information is available is a natural extension.

FieldFormer also has practical advantages for real-world sensing systems. Its mesh-free formulation supports irregular sensors, multi-resolution queries, and evolving sensor networks without requiring grid redesign or graph reconstruction. This is important in environmental monitoring, atmospheric sensing, pollution tracking, traffic infrastructure, industrial IoT, and scientific sensor deployments, where sensors may fail, move, or be added over time. In such settings, the ability to update the observational support while retaining the same coordinate-based model structure is a useful property for continual monitoring.

At the same time, several limitations remain. First, FieldFormer relies on local observability: when relevant local domains of dependence are not covered by the sensor network, reconstruction becomes underconstrained. Second, physics-based regularization remains delicate under sparse observations, especially when forcing terms, boundary conditions, or physical parameters are uncertain. Finally, FieldFormer does not eliminate the fundamental non-identifiability of global reconstruction under extreme sparse sensing; it provides a structured surrogate model whose reliability depends on sensor coverage and inductive-bias alignment.

These limitations suggest several directions for future work. Uncertainty-aware reconstruction could help distinguish well-supported sensor-space predictions from weakly constrained global completions. Adaptive sensor placement could improve observability by targeting regions where local dependencies are currently missed. Hybrid local-global architectures may combine FieldFormer's locality-aware inference with global latent priors, improving performance in regimes where local support is incomplete. Finally, more robust physics integration is needed to use partial governing knowledge without forcing models toward biased or overconstrained solutions.

Overall, FieldFormer provides both a practical method for sparse sensor-space imputation and a conceptual perspective on spatio-temporal reconstruction under sparse observability. The key lesson is that physical field reconstruction from sparse sensors is not merely an architectural problem; it is an information-constrained inference problem. By aligning model structure with the localized observational support of real sensor networks, FieldFormer offers a scalable and effective approach for constructing useful surrogate fields under severe sensing limitations.

## Broader Impact

Sparse sensor networks are widely used in environmental monitoring, atmospheric sensing, pollution tracking, traffic systems, industrial IoT, and scientific field studies, where dense measurement is often costly or infeasible. FieldFormer can improve the utility of such networks by imputing missing measurements and constructing useful surrogate fields from sparse, noisy observations. At the same time, our results emphasize that global reconstruction under extreme sparsity is not identifiable in general; model outputs should therefore be interpreted as prior-guided estimates rather than ground truth away from sensor support. Responsible deployment requires validation against held-out sensors where possible, uncertainty-aware reporting, and clear communication of where predictions are observationally supported versus weakly constrained.

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

## A  Boundary Loss Implementation

We explicitly handle boundaries: (i) **Periodic domains (Heat, SWE)**: soft equality between opposing faces, optionally including derivatives:

$$\mathcal{L}_{\mathrm{bc}}^{\mathrm{per}} = \|u(x = 0, y, t) - u(x = L_x, y, t)\|_2^2.$$

(ii) **Open domains (Pollution)**: a sponge rim loss damps edges, and an Orlanski-style radiation condition enforces

$$u_t + c_{\mathrm{eff}}\, \partial_n u \approx 0,$$

with $c_{\mathrm{eff}}$ estimated from gradients and clamped for stability. Both losses are formulated in normalized coordinates and evaluated via autograd.

## B  Baseline Training and Evaluation Protocol

All methods are trained and evaluated on the same sparse observation arrays and the same train/validation/test splits. In the standard sensor-space setting, splits are over observed sensor-time entries with an 80/10/10 ratio. In sensor-holdout experiments, validation and test sensors are removed entirely from the training context. Checkpoints are selected by validation RMSE, and final sparse metrics are computed with the same RMSE/MAE evaluator on the held-out test entries. Table 8 summarizes the model-specific input representation, objective, masking protocol, tuning choices, and output interface used in the implementation.

## C  Inference Efficiency Benchmark

We benchmark inference efficiency as wall-clock prediction time per sparse-test query using the same trained checkpoints and sparse-test splits used for the RMSE/MAE evaluation. The reported *ms/query* measures the timed prediction loop after model loading and adapter construction. We report setup time separately because methods differ in how much computation they perform before query-time prediction: for example, fixed-node imputers may cache sensor-time predictions, while grid- or set-based methods construct different intermediate representations. Peak GPU memory is measured during the timed prediction pass. These measurements are hardware- and implementation-dependent, and should be read as runtime diagnostics rather than primary model-quality metrics.

Tables 9 and 10 summarize the timing results. Direct coordinate networks such as Fourier-MLP and SIREN have very low latency because prediction is a single coordinate-function evaluation, but they are not context-conditioned transformer reconstructions. Among comparable context-conditioned methods, FieldFormer is substantially faster than Senseiver and RecFNO: its fixed-size local transformer avoids global set encoding and grid-wide reconstruction while retaining mesh-free coordinate queries. ImputeFormer has very low query-time lookup after setup because it operates over a fixed deployed node-time array and caches predictions; this should be interpreted as a fixed-node/transductive lookup regime rather than arbitrary coordinate reconstruction.

## D  Formal Definitions and Assumptions

**Definition D.1** (Finite-order local differential operator)**.** Let $u : \mathbb{R}^{d+1} \to \mathbb{R}^m$. A differential operator $\mathcal{L}$ is called a *finite-order local operator of order $r$* if

$$\mathcal{L}u(z) \;=\; F\big(z, \{D^\alpha u(z) : |\alpha| \le r\}\big), \qquad z \in \mathbb{R}^{d+1},$$

for some continuous function $F$, where $\alpha$ is a multi-index.

- **Finite-order:** Only derivatives of $u$ up to order $r < \infty$ appear.

Table 8: Training and evaluation protocol for FieldFormer and comparison methods. The comparison uses common sparse observations, splits, validation-RMSE checkpointing, and RMSE/MAE evaluation, while allowing each method to use its native input representation and objective.

| Method | Input representation | Objective/loss | Masking and context protocol | Checkpoint and evaluator | Main tuning choices | Output interface |
|---|---|---|---|---|---|---|
| FieldFormer | Query-specific local sensor-time context: relative offsets, observed values, and learned velocity-scaled geometry. | Sensor MSE on observed tuples, with optional PDE and boundary losses on synthetic PDE datasets when enabled. | Split-aware neighbor indexer restricts context to training observations and excludes the queried target from its own context. | Best validation RMSE; common sparse RMSE/MAE evaluator, with dense synthetic evaluation where reference fields exist. | Number of neighbors, temporal radius, transformer width/heads/layers, learning rate, and physics-loss weights. | Mesh-free coordinate query using local sensor context; continuous queries gather a local neighborhood from the training observations. |
| Fourier-MLP | Coordinate triples $(x, y, t)$ encoded with Fourier features. | Masked MSE on observed sensor-time targets. | Train/val/test observed-index split; no target value is used as input. | Best validation RMSE; common RMSE/MAE evaluator. | Network width/depth, Fourier frequency bands, learning rate, weight decay, and optional value/coordinate normalization. | Direct continuous coordinate function evaluated at arbitrary query coordinates. |
| Fourier-MLP Ensemble | Same coordinate Fourier-feature representation as Fourier-MLP. | Masked MSE on observed sensor-time targets for each member. | Five independently seeded data-only Fourier-MLP members trained with the same split seed and observed-index split. | Each member uses best validation RMSE checkpointing; the ensemble mean is passed to the common RMSE/MAE evaluator. | Same neural-field hyperparameters as Fourier-MLP, with independent random seeds and fixed member count. | Direct continuous coordinate query by averaging member predictions. |
| SIREN | Coordinate triples $(x, y, t)$ passed through sinusoidal layers. | Masked MSE on observed sensor-time targets. | Train/val/test observed-index split; no target value is used as input. | Best validation RMSE; common RMSE/MAE evaluator. | Network width/depth, sinusoidal frequency parameters, learning rate, weight decay, and optional normalization. | Direct continuous coordinate function evaluated at arbitrary query coordinates. |
| Fourier-MLP-PINN / SIREN-PINN | Same coordinate representation as the corresponding neural field baseline. | Masked sensor MSE plus synthetic-dataset physics residual and boundary/radiation losses where the PDE is available. | Same observed-index split; physics collocation samples are drawn from the domain and do not use full-field targets. | Best validation RMSE on sparse validation observations; common RMSE/MAE evaluator. | Neural-field hyperparameters plus residual, boundary, sponge, and radiation-loss weights and ramp schedules. | Direct continuous coordinate function; physics terms only affect training. |
| SVGP | Normalized coordinate triples with inducing points and dataset-specific kernels. | Negative variational evidence lower bound with Gaussian likelihood. | Same observed-index split; multitask runs require valid observed channels for the supervised tuple. | Best validation RMSE computed from the posterior predictive mean; common point-prediction RMSE/MAE evaluator. | Number of inducing points, kernel length-scales/output scale, likelihood noise, learning rates, and weight decay. | Probabilistic coordinate model, evaluated in the tables through its posterior predictive mean. |
| RecFNO | Sparse sensor values rasterized onto the simulation grid, with binary observation masks and coordinate channels. | Masked MSE at held-out sensor-grid targets. | Context grid is built only from training observations; the queried target is removed from the input grid during training. | Best validation RMSE; common sparse and synthetic full-field RMSE/MAE evaluator. | Fourier modes, channel width, learning rate, weight decay, batch size, and optional value normalization. | Grid-valued field prediction; arbitrary coordinates are evaluated by nearest grid location. |
| ImputeFormer | Fixed sensor-node by time-window tensor with value channels and observation masks. | Masked MSE on randomly dropped training entries within temporal windows. | Random masking within training windows; evaluation conditions on training entries only. The model spans the full deployed node set with a learnable per-node embedding for every node (held-out nodes included); held-out values are masked in the input and excluded from the loss, so no target values leak. | Best validation RMSE over held-out entries/windows; common RMSE/MAE evaluator. | Window length/stride, random mask rate, embedding sizes, layers, temporal heads, learning rate, and dropout. | Fixed-node imputation over the deployed sensor array; continuous queries are not native and are mapped to the nearest node/time when needed. |
| Senseiver | Sensor-set tokens containing normalized coordinates, time, values, and masks, plus coordinate query tokens. | Masked MSE on held-out sensor-time targets. | Sensor tokens are formed from training observations only; query targets are not included in the context. | Best validation RMSE; common RMSE/MAE evaluator. | Number of latents, latent width, cross/self attention layers and heads, learning rate, weight decay, and dropout. | Global sensor-set encoder with query decoder; supports coordinate queries through query tokens and a globally compressed sensor context. |

Table 9: Inference timing on synthetic PDE benchmarks. Times are measured on sparse-test prediction using trained checkpoints. The ms/query column excludes model loading and adapter setup; setup time is reported separately. Peak memory is measured during the timed prediction pass.

| Dataset | Model | ms/query | queries/s | Peak GPU MB | Setup s |
|---------|-------|----------|-----------|-------------|---------|
| Heat | FieldFormer | 0.034 | 2.96e+04 | 2.35e+03 | 0.494 |
| | Fourier-MLP | 4.59e-04 | 2.18e+06 | 22.6 | 0.076 |
| | Fourier-MLP Ensemble | 7.41e-04 | 1.35e+06 | 27.0 | 0.058 |
| | Fourier-MLP-PINN | 8.49e-05 | 1.18e+07 | 22.6 | 0.102 |
| | SIREN | 5.53e-05 | 1.81e+07 | 24.3 | 0.069 |
| | SIREN-PINN | 5.58e-05 | 1.79e+07 | 24.3 | 0.068 |
| | SVGP | 0.0025 | 3.96e+05 | 103 | 5.08 |
| | RecFNO | 0.330 | 3.03e+03 | 1.55e+04 | 0.171 |
| | ImputeFormer | 9.27e-06 | 1.08e+08 | 25.5 | 16.6 |
| | Senseiver | 3.89 | 257 | 13.9 | 0.206 |
| Pollution | FieldFormer | 0.034 | 2.98e+04 | 2.35e+03 | 0.626 |
| | Fourier-MLP | 7.92e-05 | 1.26e+07 | 24.9 | 0.088 |
| | Fourier-MLP Ensemble | 5.78e-04 | 1.73e+06 | 29.3 | 0.055 |
| | Fourier-MLP-PINN | 8.31e-05 | 1.20e+07 | 24.9 | 0.074 |
| | SIREN | 5.35e-05 | 1.87e+07 | 26.1 | 0.074 |
| | SIREN-PINN | 5.46e-05 | 1.83e+07 | 26.1 | 0.079 |
| | SVGP | 0.0024 | 4.17e+05 | 57.2 | 5.77 |
| | RecFNO | 0.252 | 3.98e+03 | 6.10e+03 | 0.234 |
| | ImputeFormer | 9.09e-06 | 1.10e+08 | 23.6 | 15.2 |
| | Senseiver | 4.00 | 250 | 16.2 | 0.270 |
| SWE | FieldFormer | 0.035 | 2.89e+04 | 2.35e+03 | 0.425 |
| | Fourier-MLP | 8.38e-05 | 1.19e+07 | 22.6 | 0.090 |
| | Fourier-MLP Ensemble | 7.35e-04 | 1.36e+06 | 27.2 | 0.064 |
| | Fourier-MLP-PINN | 8.37e-05 | 1.19e+07 | 22.6 | 0.096 |
| | SIREN | 2.00e-04 | 4.99e+06 | 24.3 | 0.079 |
| | SIREN-PINN | 5.22e-05 | 1.92e+07 | 24.3 | 0.060 |
| | SVGP | 0.0055 | 1.83e+05 | 289 | 6.17 |
| | RecFNO | 0.338 | 2.95e+03 | 1.56e+04 | 0.201 |
| | ImputeFormer | 8.80e-06 | 1.14e+08 | 30.4 | 16.4 |
| | Senseiver | 3.91 | 256 | 22.4 | 0.221 |

Table 10: Inference timing on real-world sparse sensor benchmarks and sensor-holdout splits. Times are measured on sparse-test prediction using trained checkpoints. The ms/query column excludes model loading and adapter setup; setup time is reported separately. Peak memory is measured during the timed prediction pass.

| Dataset | Model | ms/query | queries/s | Peak GPU MB | Setup s |
|---|---|---|---|---|---|
| Atmospheric | FieldFormer | 0.035 | 2.82e+04 | 2.39e+03 | 0.519 |
| | Fourier-MLP | 2.58e-04 | 3.88e+06 | 38.4 | 0.041 |
| | Fourier-MLP Ensemble | 8.30e-04 | 1.21e+06 | 43.0 | 0.062 |
| | SIREN | 2.02e-04 | 4.95e+06 | 40.0 | 0.050 |
| | SVGP | 0.0049 | 2.03e+05 | 203 | 6.25 |
| | RecFNO | 0.254 | 3.94e+03 | 4.42e+03 | 0.193 |
| | ImputeFormer | 8.42e-06 | 1.19e+08 | 41.4 | 15.4 |
| | Senseiver | 4.37 | 229 | 39.8 | 0.270 |
| Pollution | FieldFormer | 0.034 | 2.95e+04 | 2.37e+03 | 0.518 |
| | Fourier-MLP | 1.36e-04 | 7.37e+06 | 33.0 | 0.065 |
| | Fourier-MLP Ensemble | 6.68e-04 | 1.50e+06 | 37.4 | 0.061 |
| | SIREN | 6.81e-05 | 1.47e+07 | 34.6 | 0.063 |
| | SVGP | 0.0028 | 3.61e+05 | 110 | 5.05 |
| | RecFNO | 0.250 | 4.00e+03 | 4.36e+03 | 0.274 |
| | ImputeFormer | 1.98e-05 | 5.04e+07 | 30.6 | 16.8 |
| | Senseiver | 4.56 | 220 | 29.0 | 0.238 |
| Atmospheric-split | FieldFormer | 0.035 | 2.89e+04 | 2.39e+03 | 0.578 |
| | Fourier-MLP | 2.45e-04 | 4.08e+06 | 38.4 | 0.046 |
| | Fourier-MLP Ensemble | 8.21e-04 | 1.22e+06 | 43.0 | 0.085 |
| | SIREN | 2.02e-04 | 4.95e+06 | 40.0 | 0.042 |
| | SVGP | 0.0050 | 2.01e+05 | 203 | 5.68 |
| | RecFNO | 0.275 | 3.64e+03 | 4.81e+03 | 0.150 |
| | ImputeFormer | 8.33e-05 | 1.20e+07 | 41.4 | 16.2 |
| | Senseiver | 4.66 | 214 | 39.8 | 0.232 |
| Pollution-split | FieldFormer | 0.034 | 2.91e+04 | 2.37e+03 | 0.525 |
| | Fourier-MLP | 2.62e-04 | 3.81e+06 | 33.0 | 0.053 |
| | Fourier-MLP Ensemble | 8.38e-04 | 1.19e+06 | 37.4 | 0.089 |
| | SIREN | 2.19e-04 | 4.57e+06 | 34.6 | 0.038 |
| | SVGP | 0.0028 | 3.55e+05 | 110 | 5.58 |
| | RecFNO | 0.272 | 3.68e+03 | 4.74e+03 | 0.096 |
| | ImputeFormer | 5.90e-05 | 1.70e+07 | 30.6 | 18.0 |
| | Senseiver | 4.81 | 208 | 29.0 | 0.223 |

- **Local:** $\mathcal{L}u(z)$ depends only on $u$ and its derivatives evaluated at the same point $z$, not on integrals or values of $u$ away from $z$.

Examples include the heat operator $\partial_t - \Delta$, the wave operator $\partial_{tt} - c^2\Delta$, and the advection operator $\partial_t + v \cdot \nabla$. Nonlocal operators such as the fractional Laplacian $(-\Delta)^{\alpha/2}$ are excluded.

**Assumption D.2. Locality Assumption:** Assume the setup of Theorem 5.1 with stencil $\mathcal{S} \subset \{-s_{\max}, \ldots, s_{\max}\}^d \times \{0, \ldots, k\}$ of size $N := |\mathcal{S}|$. Let the explicit update at $(\mathbf{i}, n+1)$ be

$$U^{n+1}(\mathbf{i}) \;=\; \psi\big(\{U^{n-\ell}(\mathbf{i}+\mathbf{s}) : (\mathbf{s}, \ell) \in \mathcal{S}\}\big),$$

$$\psi : (\mathbb{R}^q)^N \to \mathbb{R}^q.$$

**Assumption D.3. Sampling Model:** Let the stencil factor as

$$\mathcal{S} = \mathcal{S}_x \times \mathcal{T}, \qquad \mathcal{S}_x \subset \{-s_{\max}, \ldots, s_{\max}\}^d,$$

$$|\mathcal{S}_x| = N_x, \qquad \mathcal{T} = \{0, 1, \ldots, k\}.$$

Thus the stencil size is $N = N_x(k+1)$, but coverage is determined entirely by the $N_x$ spatial indices. A sensor placed at spatial offset $s \in \mathcal{S}_x$ provides all $k+1$ time levels.

We randomly place $m_x$ sensors *uniformly without replacement* among the $N_x$ essential spatial locations, independently of the stencil values. FieldFormer then takes all available measurements.

**Assumption D.4. Missed-Support Ambiguity Margin:** Let $V \sim \mathcal{D}$ be the random input (stencil values), and $\psi : (\mathbb{R}^q)^N \to \mathbb{R}^q$ the local update map. Index the stencil as $\mathcal{S} = \mathcal{S}_x \times \mathcal{T}$ with $\mathcal{S}_x \subset \{-s_{\max}, \ldots, s_{\max}\}^d$ and $\mathcal{T} = \{0, \ldots, k\}$, so $N = |\mathcal{S}| = N_x(k+1)$ with $N_x = |\mathcal{S}_x|$.

Assume there exist nonnegative weights $(a_{(s,\ell)})_{(s,\ell) \in \mathcal{S}}$ with

$$\sum_{(s,\ell) \in \mathcal{S}} a_{(s,\ell)} = 1.$$

For longitudinal sampling (a spatial miss hides all time levels), aggregate the weights per spatial offset

$$\bar{a}_s \;:=\; \sum_{\ell \in \mathcal{T}} a_{(s,\ell)}, \qquad s \in \mathcal{S}_x,$$

so that $\bar{a}_s \geq 0$ and $\sum_{s \in \mathcal{S}_x} \bar{a}_s = 1$. For a subset $J \subseteq \mathcal{S}_x$, define its *influence mass* as $A(J) := \sum_{s \in J} \bar{a}_s$.

Assume there exists a constant $c_{\mathrm{amb}} > 0$ such that, for every missed spatial set $J \subseteq \mathcal{S}_x$, there is a transformation $T_J$ that acts only on $J \times \mathcal{T}$ and leaves all coordinates in $(\mathcal{S}_x \setminus J) \times \mathcal{T}$ unchanged. Thus $V$ and $T_J V$ are observationally indistinguishable to any estimator that only observes $(\mathcal{S}_x \setminus J) \times \mathcal{T}$. The two completions have separated updates in proportion to the missed influence mass:

$$\mathbb{E}\big[\|\psi(V) - \psi(T_J V)\|_2\big] \;\geq\; c_{\mathrm{amb}}\, A(J), \tag{3}$$

where the expectation is over $V \sim \mathcal{D}$ and any internal randomness of $T_J$.

*Remark* D.5 (Scope of the ambiguity margin). Assumption D.4 is a worst-case *non-identifiability* condition, not an average-case guarantee: it asserts the *existence* of one separated, observationally indistinguishable completion, which is exactly what a two-world minimax bound requires. It is non-vacuous when the missed spatial support is non-redundant for the local operator—most cleanly for hyperbolic/advective stencils (upwind transport, SWE), where the finite domain of dependence makes upstream points irreducible, and for parabolic stencils (Heat) at the one-step update-map level, where the discrete Laplacian has nonzero coefficients on each spatial neighbor. The margin degrades ($c_{\mathrm{amb}} \to 0$) when strong smoothness, low-rank structure, or spatial correlation renders the missed values recoverable from the observed coordinates, or for degenerate/symmetric configurations. The constant $c_{\mathrm{amb}}$ absorbs the stencil-coefficient magnitudes on the missed offsets and the spread of admissible perturbations there.

# E    Proofs of Theorems

## E.1    Proof of Theorem 5.1

**Theorem E.1** (Expressivity for Finite-Stencil Local Dynamics). *Let $u : \mathbb{R}^d \times \mathbb{R} \to \mathbb{R}^q$ solve a parabolic or hyperbolic PDE*

$$F\big(z, u(z), \nabla u(z), \dots, D^r u(z)\big) = 0, \qquad z = (x, t),$$

*with a finite-order, local operator. Assume a consistent* explicit *finite-difference scheme with compact stencil $\mathcal{S} \subset \{-s_{\max}, \dots, s_{\max}\}^d \times \{0, \dots, k\}$ and a CFL-admissible time step $\Delta t$. Let $U_h$ denote the numerical solution generated by this scheme, and write its local update as*

$$U_h^{n+1}(\mathbf{i}) = \psi(v_{\mathbf{i},n}), \qquad v_{\mathbf{i},n} = \big(U_h^{n-\ell_j}(\mathbf{i} + \mathbf{s}_j)\big)_{j=1}^N,$$

*where $\{(\mathbf{s}_j, \ell_j)\}_{j=1}^N$ is an ordering of $\mathcal{S}$ and $N = |\mathcal{S}|$. For any compact $K \subset \mathbb{R}^{d+1}$, define the compact set of stencil-value tuples*

$$\mathcal{V}_K := \big\{ v_{\mathbf{i},n} \in (\mathbb{R}^q)^N : (x_{\mathbf{i}}, t_{n+1}) \in K \big\}.$$

*Then for any $\varepsilon > 0$ there exists a FieldFormer with neighborhood size $m \geq |\mathcal{S}|$, hidden width $d'$, and depth $L$ such that*

$$\sup_{v \in \mathcal{V}_K} \|\psi(v) - \mathrm{FF}_\theta(E(v))\|_2 < \varepsilon,$$

*where $E(v)$ denotes the FieldFormer tokenization of the ordered stencil tuple $v$.*

*Proof.* Let $u : \mathbb{R}^d \times \mathbb{R} \to \mathbb{R}^q$ solve a parabolic or hyperbolic PDE

$$F\big(z, u(z), \nabla u(z), \dots, D^r u(z)\big) = 0, \qquad z = (x, t),$$

whose differential operator is finite-order and local (Def. 3.1). Assume a consistent explicit finite-difference scheme with compact stencil $\mathcal{S} \subset \{-s_{\max}, \dots, s_{\max}\}^d \times \{0, \dots, k\}$ and CFL-admissible $\Delta t > 0$. Let $U_h$ denote the numerical solution produced by this scheme.

**1) Discrete locality and finite map.**    On a uniform grid with spatial step $h > 0$ and time step $\Delta t > 0$, the update rule at index-time $(\mathbf{i}, n+1)$ is given by

$$U_h^{n+1}(\mathbf{i}) = \psi(v_{\mathbf{i},n}), \qquad v_{\mathbf{i},n} = \big(U_h^{n-\ell_j}(\mathbf{i} + \mathbf{s}_j)\big)_{j=1}^N,$$

for a continuous function $\psi : (\mathbb{R}^q)^N \to \mathbb{R}^q$, where $\{(\mathbf{s}_j, \ell_j)\}_{j=1}^N$ is an ordering of $\mathcal{S}$ and $N := |\mathcal{S}|$. Continuity follows because $\psi$ consists of finitely many arithmetic operations on the stencil entries and coefficients of the PDE discretization.

**2) Tokenization of the stencil.**    A FieldFormer uses $m$ tokens ($m \geq N$); the first $N$ encode the stencil values and their relative coordinates. Each token concatenates (i) relative position $(h\,\mathbf{s}_j, \Delta t\,\ell_j)$ and (ii) the field value $U_h^{n-\ell_j}(\mathbf{i} + \mathbf{s}_j)$, producing an input vector $x \in \mathbb{R}^{md_{\mathrm{in}}}$. If $m > N$, padded tokens are masked so they have no effect. Hence there exists a continuous encoder $E : (\mathbb{R}^q)^N \to \mathbb{R}^{md_{\mathrm{in}}}$. Because the ordering and relative offsets are fixed and the stencil values are included in the tokens, $E$ is injective on the set of stencil tuples considered below.

**3) Compact domain of interest.**    Fix a compact domain $K \subset \mathbb{R}^{d+1}$. The set of all stencil value tuples attained over $K$,

$$\mathcal{V}_K := \Big\{ \big(U_h^{n-\ell_j}(\mathbf{i} + \mathbf{s}_j)\big)_{j=1}^N : (x_{\mathbf{i}}, t_{n+1}) \in K \Big\}, \qquad (x_{\mathbf{i}}, t_n) = (\mathbf{i}h, n\Delta t).$$

is a finite, hence compact, subset of $(\mathbb{R}^q)^N$. Since $E$ is continuous and injective on $\mathcal{V}_K$, it has a continuous inverse on its compact image $E(\mathcal{V}_K)$. Therefore

$$\phi(x) := \psi(E^{-1}(x))$$

is continuous on $E(\mathcal{V}_K)$.

**4) Universal approximation by the FieldFormer.** A FieldFormer encoder followed by pooling defines a continuous permutation-invariant set function of the token multiset. This does not lose the stencil ordering needed by $\psi$: the relative offsets $(h\mathbf{s}_j, \Delta t\,\ell_j)$ are distinct fixed identifiers included in the tokens, so any permutation-invariant function of the position-value tokens can represent a function of the corresponding ordered stencil tuple. Equivalently, the Deep Sets representation theorem (Zaheer et al., 2017) applies to the token set, while the injective tokenization above recovers the ordered tuple on $\mathcal{V}_K$; this is also consistent with transformer universality results with positional encodings (Yun et al., 2019), since the relative coordinates play the role of token identifiers. Thus the FieldFormer/MLP stack defines a continuous function $g_\theta : \mathbb{R}^{md_{\mathrm{in}}} \to \mathbb{R}^q$. By universal approximation for MLPs (Hornik, 1991), for any $\varepsilon > 0$ there exist width $d'$ and depth $L$ such that

$$\sup_{x \in E(\mathcal{V}_K)} \| g_\theta(x) - \phi(x) \|_2 < \varepsilon.$$

Writing $\mathrm{FF}_\theta = g_\theta$ and substituting $x = E(v)$ gives

$$\sup_{v \in \mathcal{V}_K} \| \mathrm{FF}_\theta(E(v)) - \psi(v) \|_2 < \varepsilon.$$

Hence for some $m \geq |\mathcal{S}|$, width $d'$, and depth $L$, FieldFormer uniformly approximates the finite-stencil update map on $\mathcal{V}_K$, completing the proof. □

## E.2 Proof of Theorem 5.3

**Theorem E.2** (Missed-Support Non-Identifiability Under Longitudinal Sampling). *Sample $m_x$ spatial locations uniformly without replacement from $\mathcal{S}_x$, let $I \subseteq \mathcal{S}_x$ be the sensed set and $I^c$ the missed set. Under equation equation 3, let $\hat{u}$ be any estimator using only the observed stencil block $V_{I \times \mathcal{T}}$. For fixed $I$, define the two completion risks*

$$\mathrm{risk}_I(V) := \mathbb{E}[\|\hat{u}(V_{I \times \mathcal{T}}) - \psi(V)\|_2 \mid I],$$

*and*

$$\mathrm{risk}_I(T_{I^c}V) := \mathbb{E}[\|\hat{u}(V_{I \times \mathcal{T}}) - \psi(T_{I^c}V)\|_2 \mid I],$$

*where $T_{I^c}V$ agrees with $V$ on the observed coordinates and differs only on the missed spatial support. Then there exists a constant $c > 0$ such that*

$$\inf_{\hat{u}} \mathbb{E}_I [\max\{\mathrm{risk}_I(V), \mathrm{risk}_I(T_{I^c}V)\}] \geq c\,\mathbb{E}_I[A(I^c)] = c\left(1 - \frac{m_x}{N_x}\right).$$

*Proof.* Let $V \in (\mathbb{R}^q)^N$ denote the random tuple of stencil values under distribution $\mathcal{D}$, with $\mathcal{S} = \mathcal{S}_x \times \mathcal{T}$ and $N = |\mathcal{S}|$. The local update target is $\psi(V) \in \mathbb{R}^q$.

**1) Conditioning on sensed sites.** Fix a sampled set $I \subseteq \mathcal{S}_x$ of $m_x$ observed spatial offsets and write $J = I^c$ for the missed set. For each $s \in I$, all $k+1$ time coordinates $(s, \ell)$ are known, while coordinates with $s \in J$ are unobserved. Any estimator using all available data must be a function of the observed block, $\hat{u} = \hat{u}(V_{I \times \mathcal{T}})$.

**2) Two indistinguishable completions.** By Assumption D.4, there exists a set-level transformation $T_J$ that acts only on $J \times \mathcal{T}$ and leaves $I \times \mathcal{T}$ unchanged. Hence

$$(T_J V)_{I \times \mathcal{T}} = V_{I \times \mathcal{T}},$$

so the estimator receives the same input for the two completions $V$ and $T_J V$. Applying the triangle inequality gives

$$\|\psi(V) - \psi(T_J V)\|_2 \leq \|\hat{u}(V_{I \times \mathcal{T}}) - \psi(V)\|_2 + \|\hat{u}(V_{I \times \mathcal{T}}) - \psi(T_J V)\|_2.$$

Taking expectation over $V$ and any internal randomness of $T_J$, conditional on $I$, yields

$$\mathrm{risk}_I(V) + \mathrm{risk}_I(T_J V) \geq \mathbb{E}[\|\psi(V) - \psi(T_J V)\|_2 \mid I].$$

Therefore,

$$\max\{\mathrm{risk}_I(V), \mathrm{risk}_I(T_J V)\} \geq \frac{1}{2} \mathbb{E}[\|\psi(V) - \psi(T_J V)\|_2 \mid I].$$

**3) Applying the missed-support ambiguity margin.** Using equation 3 with $J = I^c$,

$$\mathbb{E}[\|\psi(V) - \psi(T_{I^c}V)\|_2 \mid I] \geq c_{\text{amb}} A(I^c).$$

Thus, for $c = c_{\text{amb}}/2$,

$$\max\{\text{risk}_I(V), \text{risk}_I(T_{I^c}V)\} \geq c A(I^c).$$

**4) Averaging over random sampling.** For uniform $m_x$-of-$N_x$ sampling without replacement, each spatial site is missed with probability $1 - m_x/N_x$, so

$$\mathbb{E}_I\big[A(I^c)\big] = \sum_{s \in \mathcal{S}_x} \bar{a}_s \, \Pr(s \notin I) = \Big(1 - \frac{m_x}{N_x}\Big)\sum_s \bar{a}_s = 1 - \frac{m_x}{N_x}.$$

Taking expectation over $I$ and then the infimum over estimators yields

$$\inf_{\hat{u}} \ \mathbb{E}_I\left[\max\{\text{risk}_I(V), \text{risk}_I(T_{I^c}V)\}\right] \geq c \, \mathbb{E}_I\big[A(I^c)\big] = c\Big(1 - \frac{m_x}{N_x}\Big),$$

as claimed. $\qquad\qquad\qquad\qquad\qquad\qquad\qquad\qquad\qquad\qquad\qquad\qquad\qquad\qquad\qquad\qquad\qquad\quad \square$

## F   Additional Results

Table 11 reports sensor-holdout results, used as a proxy for spatial generalization when dense full-field ground truth is unavailable. FieldFormer is stronger on most atmospheric holdout variables, while Gamma Field improves $PM_{10}$ holdout performance on the pollution dataset. The mixed results indicate that local coordinate-dependent scaling is useful in some heterogeneous regimes, but is not a universal substitute for observational support at unseen sensors.

Table 11: Ablation study on real-world benchmarks: sensor-holdout evaluation. Results are reported as mean ± std over bootstrap samples. Lower is better.

| Dataset | Variable | FieldFormer RMSE | FieldFormer MAE | Gamma Field RMSE | Gamma Field MAE |
|---|---|---|---|---|---|
| Atmospheric | AT | **4.63 ± 0.6376** | **3.14 ± 0.4384** | 4.683 ± 0.6321 | 3.201 ± 0.46 |
| Atmospheric | RH | 15.75 ± 2.517 | **10.04 ± 1.568** | **15.63 ± 1.952** | 10.11 ± 1.318 |
| Atmospheric | $U_x$ | **2.26 ± 0.6081** | **1.282 ± 0.2269** | 2.362 ± 0.6137 | 1.33 ± 0.2493 |
| Atmospheric | $V_y$ | **1.502 ± 0.1426** | **1.026 ± 0.09529** | 1.671 ± 0.1498 | 1.122 ± 0.1189 |
| Pollution | $PM_{10}$ | 115 ± 13.11 | 69.63 ± 9.17 | **92.63 ± 9.512** | **56.73 ± 6.764** |
| Pollution | $PM_{2.5}$ | **45.95 ± 4.77** | **26.76 ± 3.152** | 47.5 ± 4.847 | 27.14 ± 3.161 |

