# OpenReview forum: "FieldFormer: Locality-Aware Transformers for Spatio-Temporal Modeling on Sparse Sensor Networks"
_TMLR — Under review for TMLR_

### Review · Reviewer_jzvM · 2026-06-19

**Summary Of Contributions:**

- The paper studies spatio-temporal reconstruction from persistent sparse sensor networks—settings in which observations are dense in time but limited to a small fixed set of spatial locations.
- It introduces a conceptual distinction between sensor-space modeling (predicting at observed sensor locations) and unconstrained global field reconstruction, and argues that the former is the more identifiable objective under extreme sparsity.
- It proposes FieldFormer, a mesh-free architecture that constructs an adaptive local neighborhood around each query via a learnable velocity-scaled distance metric, encodes relative offsets and observations as tokens, and aggregates them with a local transformer encoder. Two variants are also introduced: FieldFormer-PINN (with differentiable PDE-residual regularization) and FieldFormer–Gamma Field (with a coordinate-dependent metric).
- Two theoretical results are presented: Theorem 5.1 (expressivity for locally structured spatio temporal dynamics) and Theorem 5.3 (average-case lower bound under longitudinal sampling).
- Empirically, FieldFormer is compared against eight baselines across three synthetic PDE benchmarks and two real-world datasets, evaluated under three regimes: sensor-space imputation, sensor-holdout generalization, and global field reconstruction. Ablations isolate the contributions of the PINN variant, the Gamma Field variant, and the local transformer (vs an MLP variant).
- An anonymized code repository is provided for review.

**Audience:**

Yes

**Audience Explanation:**

- Sparse spatio-temporal reconstruction is a substantive problem for TMLR’s audience, with active research communities in spatial statistics, geostatistics, environmental ML, physics-informed learning, and operator learning. The mesh-free design and the explicit attention to evolving sensor topologies have practical appeal beyond the specific datasets used.
- Even with the concerns above, the conceptual question the paper raises—how to think about reconstruction quality when the observation geometry itself is the bottleneck—is one the community should engage with. A revised version that addresses the comparison gaps and reporting issues would be of clear interest to readers working on PDE learning, neural fields, and applied sensor data.

**Broader Impact Concerns:**

The paper includes a Broader Impact section that appropriately notes (i) the wide deployment of sparse sensor networks in environmental, infrastructure, and scientific contexts, (ii) that global reconstruction under extreme sparsity is generally not identifiable, and (iii) that outputs should be interpreted as prior-guided estimates rather than ground truth far from sensor support. This framing is reasonable and largely sufficient.

**Claims And Evidence:**

Yes

**Claims Explanation:**

Several core claims are not adequately supported by the evidence presented. These issues are recoverable through revision.

1. **False dichotomy in the conceptual framing.** The paper contrasts sensor-space modeling against unconstrained global reconstruction and concludes the former is the right objective. This ignores the standard alternative for under-determined problems: methods that genuinely retain multiple plausible solutions (e.g., CSDI, ConvCNP/ConvNP, Neural Processes, deep ensembles evaluated by predictive distribution). The only baseline labeled probabilistic (SVGP) is, in practice, a variationally-optimized single-solution method evaluated by its posterior mean—it does not represent the class of uncertainty-expressive methods that the paper’s identifiability argument calls for. The paper’s “we included a probabilistic baseline” framing therefore rests on a category mismatch.
1. **Local decomposition framed as a new paradigm.** Operationally FieldFormer performs per-query local prediction; the repeated “globally grounded through continuous coordinates” framing is not substantiated by any ablation. No baseline isolates the contribution of the local-decomposition strategy from that of the specific architecture, so the contribution of the decomposition strategy cannot be separated from that of the specific architecture. Table 5 already shows FieldFormer-MLP within 0.3% of full FieldFormer on Heat, reinforcing this concern.
1. **Reporting does not match stated conventions.** Captions for Tables 1, 2, 4–8 state “mean ± standard deviation over bootstrap samples,” but the tables contain only point estimates. Comparisons differing by < 0.01 (e.g., Table 1 Heat RMSE) cannot be assessed for statistical significance.
1. **Experimental fairness requires more transparency.** The mesh-dependent baselines (RecFNO, ImputeFormer, Senseiver) are adapted to the sparse setting via rasterization, fixed-node windows, or global-latent bottlenecks, which may push them outside their design regimes. An explicit table comparing each method’s loss function, input representation, and hyperparameter search budget would clarify whether the comparison is like-for-like.

**Requested Changes:**

### Critical
- Add genuinely uncertainty-expressive baselines. Include at least one baseline that genuinely retains and is evaluated on its predictive distribution. SVGP under variational approximation is effectively a single-solution method and does not represent this class.
- Add ablation baselines that isolate the contribution of local decomposition from the architecture. The current results do not separate (i) the effect of restricting prediction to a local neighborhood from (ii) the effect of the specific transformer-based aggregation within that neighborhood. Table 5 (FieldFormer-MLP within 0.3% of full FieldFormer on Heat) already suggests this separation is needed. Suitable baselines are at the authors' discretion.
- Report bootstrap standard deviations on Tables 1–8, as the captions promise; tables currently show only point estimates.
- Make the like-for-like nature of the baseline comparison verifiable. Readers should be able to confirm that the compared methods are solving comparable problems with comparable tuning effort. The form of disclosure (e.g., a protocol table, expanded text, or appendix) is at the authors’ discretion.

### Important
- Substantiate scalability and efficiency claims empirically. The current claims about mesh-free efficiency rest on qualitative statements; concrete empirical comparison against the baselines is needed. Which efficiency metrics best capture the claim is at the authors’ discretion.
- Strengthen the Section 1 motivation beyond ML-internal references. The real-world stakes of persistent sparse sensor networks are currently asserted but supported only by closely-related ML papers. Citations from the application domains themselves (e.g., environmental monitoring, observation system design, IoT infrastructure) would establish the broader stakes the framing relies on. Selection of appropriate sources is at the authors’ discretion.
- Address diffusion-based imputation (CSDI and related score-based methods): include as baselines or justify the omission in Related Work.

### Minor
- Define paper-specific terminology at first use (“velocity-scaled distance metric,” “fixed maximal sparse context,” “token geometry”), or replace with standard terms. The Abstract and the Section 1 paragraph beginning “For each query location…” propagate this issue.
- Improve self-containedness around the Transformer encoder. The paper’s novelty claim is that the learned metric modulates attention, which is hard for readers outside core ML to evaluate without an explicit treatment of the attention machinery being modified. The level and placement of background material is at the authors’ discretion.
- Substantiate or remove unsupported framings. The “globally grounded through continuous coordinates” phrase is never shown to be load-bearing; either support it or rephrase. The “naturally adapts” claim in Section 4.1 is similarly qualitative and would benefit from quantitative grounding.

---

> ### Author Response · Authors · 2026-07-01
> **Response to Reviewer jzvM**
>
> Thank you for the detailed review. We address the main concerns below and will incorporate the corresponding paper revisions.
>
> ### Uncertainty-expressive baselines and framing of identifiability
>
> Our argument is not "sensor-space prediction or global reconstruction" dichotomy. The intended claim is narrower: under extreme sparse sensing, **point-estimate global reconstruction** is underidentified unless one adds strong priors or represents uncertainty over multiple plausible completions.
>
> Uncertainty-expressive methods are complementary to this framing. Our tables evaluate point predictions using RMSE/MAE against one realized target field, so distributional methods must still be reduced to a point estimate, typically a predictive mean or median. That evaluates point accuracy, not the full quality of calibrated or multimodal uncertainty.
>
> We will revise the SVGP discussion accordingly: SVGP is probabilistic, but our benchmark uses its posterior mean to match the point-prediction protocol, so it should not be presented as covering the full class of uncertainty-aware reconstruction methods. We are also running a **deep ensemble of Fourier-MLP** as an additional predictive baseline, reporting the ensemble mean under the same RMSE/MAE protocol and uncertainty diagnostics such as predictive variance or interval coverage where appropriate.
>
> ### Local decomposition versus architecture
>
> The current experiments already separate these factors more than the paper makes explicit.
>
> FieldFormer-MLP is the controlled ablation for **local decomposition without transformer aggregation**: it keeps the same query-specific local context, but replaces the transformer aggregator. Senseiver provides the complementary comparison: it is a global attention-based set model that does not use FieldFormer's query-specific local neighborhood, learned local metric, or per-query decomposition. It is not a one-switch ablation, but it is the relevant no-local-decomposition attention comparator already in the experiments.
>
> Together, FieldFormer, FieldFormer-MLP, and Senseiver compare local decomposition versus global attention, and transformer aggregation versus MLP aggregation within a local context. We will make this logic explicit and extend the FieldFormer-MLP ablation to the two real-world datasets.
>
> ### Bootstrap standard deviations in tables
>
> The reported standard deviations are **bootstrap standard deviations of the RMSE/MAE estimates**, not predictive uncertainty. They measure how much the aggregate metric changes when resampling the held-out evaluation set; they do not measure epistemic uncertainty over possible global field completions or per-query predictive variance.
>
> Small bootstrap standard deviations are expected here because the held-out sets are large and the processes are smooth or highly structured. They show metric stability under data resampling, not certainty about global reconstruction.
>
> ### Baseline comparison transparency
>
> The comparison is like-for-like at the task/evaluation level: methods use the same sparse observations, train/validation/test split, missing-entry masks, validation-RMSE checkpoint selection, and RMSE/MAE evaluator on the same held-out targets. These details are available in the anonymous repository provided with the submission.
>
> The training objectives and hyperparameters are not literally identical, because the baselines have different native forms: coordinate neural fields, grid models, fixed-node imputers, global set models, SVGP, and PINN variants require different inputs and losses. The comparison controls the data split and evaluation target, while allowing method-appropriate training. We will add a compact appendix table listing each baseline's input representation, objective, masking strategy, checkpoint metric, and main tuning choices.
>
> ### Scalability and efficiency claims
>
> We will add an inference-efficiency comparison. The main deployment cost in our setting is prediction under sparse persistent sensor networks, so we will report wall-clock prediction time per evaluated query and peak GPU memory where available, using trained checkpoints, the same evaluation script, device, and batching protocol as the RMSE/MAE results. We will frame this as empirical throughput under the studied sparse-sensing protocol, not as an architecture-intrinsic complexity theorem.
>
> ### Paper revisions for scope, related work, and exposition
>
> We will also revise the paper for scope and exposition: add application-domain references for persistent sparse sensor networks, expand related work on diffusion/score-based imputation such as CSDI, define paper-specific terms at first use, and clarify how the learned metric affects neighborhood construction and transformer aggregation.

---

> > ### Comment · Reviewer_jzvM · 2026-07-09
> >
> > Thank you for the detailed rebuttal and revision. I have verified the revised manuscript against my requested changes, and I confirm that they have been substantively addressed, including all Critical and Important items.
> >
> > One remaining request. In the rebuttal, you narrowed the central claim to point-estimate global reconstruction being underidentified, and this narrowing is consistently implemented in the evaluation methodology and interpretation. However, the manuscript's most prominent statements of the claim still assert that sensor-space modeling is "a more identifiable objective than unconstrained global field recovery" without the point-estimation scope. I noticed this in the Abstract, Section 1, and Section 5.4. As written, these restate the broader claim that my original review argued was not supported. Please ensure that the claim, wherever prominently stated, is scoped as in your rebuttal. The specific wording and placement are at the authors' discretion.
> >
> > Conditional on this, I am prepared to update my answer to "Are the claims made in the submission supported by accurate, convincing and clear evidence?" from No to Yes.

---

> ### Author Response · Authors · 2026-07-10
>
> Thank you, we appreciate you confirming that the Critical and Important items are substantively addressed, and we agree with this final scoping request.
>
> We have scoped the central claim to **point-estimate** global reconstruction wherever it is prominently stated, so it no longer reads as a claim about all global reconstruction (including uncertainty-expressive methods that retain multiple completions):
>
> - **Abstract:** now "a more identifiable objective than *point-estimate* global field recovery."
> - **Section 1:** same scoping, plus an explicit clause — "This identifiability gap concerns single deterministic (point-estimate) field predictions; methods that explicitly represent uncertainty over multiple plausible completions are a complementary route rather than a counterexample."
> - **Section 5.4:** now "a more identifiable objective than *point-estimate* global field recovery under extreme sparsity."
> - For consistency, we also scoped the same distinction in **Section 3** (Problem Formulation): "point-estimate global field reconstruction."
>
> We're grateful for the thorough review, which has clearly improved the paper's precision.

---

> > ### Comment · Reviewer_jzvM · 2026-07-10
> >
> > Thank you for the revision. I have verified the updated manuscript and confirmed that the central claim is now consistently scoped to point-estimate global field reconstruction. This resolves my remaining concern.
> > Accordingly, I have updated my answer to "Are the claims made in the submission supported by accurate, convincing and clear evidence?" from No to Yes.

---

> > > ### Comment · Action_Editor_z24K · 2026-07-10
> > > **Thank you all for the great discussion**
> > >
> > > I just wanted to step in real quickly here and thank you all for a clear and effective discussion to resolve the concerns reviewer jzvM raised.
> > >
> > > This is the right way to do peer review, thank you for your efforts and being a great example for others.

---

### Review · Reviewer_NuTG · 2026-06-22

**Summary Of Contributions:**

This paper studies spatiotemporal physical field modeling in sparse, noisy, irregular, and long-term deployed sensor networks. Its central position is that, under extreme spatial sparsity, recovering a complete global latent field from only a small number of sensor trajectories is generally non-identifiable. Therefore, a more appropriate objective is sensor-space imputation on the persistent sensor support, while interpreting predictions far from the observational support as prior-guided surrogate fields. Methodologically, the paper proposes FieldFormer: for each query, it constructs a fixed-size maximal local spatiotemporal context, rescales relative coordinate offsets using a velocity-scaled learnable metric, and aggregates neighborhood tokens with a local Transformer encoder. The paper also provides a local expressivity theorem and a miss-coverage lower bound to explain the relationship between local observational coverage and recoverability.

**Audience:**

Yes

**Audience Explanation:**

The paper studies an important problem in spatio-temporal modeling and sparse sensing. Its perspective on sensor-space imputation under extreme spatial sparsity is likely to be of interest to part of the TMLR audience.

**Broader Impact Concerns:**

I do not have major broader-impact concerns.

**Claims And Evidence:**

Yes

**Claims Explanation:**

The paper provides substantial empirical evidence through experiments on synthetic and real-world datasets, comparisons with multiple baselines, and ablation studies. While some claims could be stated more cautiously, the main conclusions are generally supported by the presented results.

**Requested Changes:**

**Weaknesses**

1. **The current theoretical proofs are insufficient to support the level of theoretical credibility that the paper attempts to establish.** My specific concerns are listed under “Questions.” The authors should carefully check whether there are errors in the formula derivations and whether there are questionable logical gaps throughout the paper.

2. **The proposed FieldFormer relies on local observational support and has structural limitations for unseen-sensor prediction and global reconstruction.** The paper’s view that “under extreme spatial sparsity, global field reconstruction is intrinsically non-identifiable, and therefore the focus should be on sensor-space imputation” is reasonable and insightful. However, this also means that FieldFormer’s advantage depends heavily on the existence of sufficient observational support near the query location. When an entire site is held out, or when predictions are required in new regions far from existing sensors, the local evidence available to FieldFormer is substantially reduced. In such cases, methods with stronger global priors may perform better. Table 4 also shows that, in the real sensor-holdout setting, ImputeFormer / Senseiver outperform FieldFormer on atmospheric AT/RH and pollution PM10/PM2.5.


**Questions**

1. Theorem 5.1 seems to claim that a continuous PDE solution $u(z)$ can be uniformly approximated on a compact domain, namely

    $$\sup_{z\in K} \\|u(z)-\hat u(z)\\| < \varepsilon.$$

    However, from the proof, what is actually established seems to be only that FieldFormer can approximate the discrete update map of a finite stencil,

    $$u^{n+1}(i)=\psi \left({u^{n-\ell}(i+s):(s,\ell)\in S}\right),$$

    that is,

    $$\sup_{v\in V_K} \\|\psi(v)-\hat\psi(v)\\| < \varepsilon.$$

    The latter does not directly imply the former. The transition from approximating a finite-stencil update rule to uniformly approximating the solution of a continuous PDE requires a standard numerical-analysis convergence argument, usually involving an error decomposition such as

    $$\\|u-\hat u \\| \le \underbrace{\\|u-u_h\\|}\_{\text{discretization error}} + \underbrace{\\|u_h-\hat u\\|}\_{\text{network approximation error}},$$

    where $u_h$ denotes the numerical solution produced by a finite-difference scheme. From the current proof, it seems that only the second term is controlled, while the first term $\\|u-u_h\\|$ is not.

    Could the authors further clarify what additional assumptions are needed to control $\\|u-u_h\\|$? If the main purpose of this result is actually to show that FieldFormer has sufficient expressivity to approximate finite-stencil dynamics, would it be more accurate to state the theorem directly as an approximation result for the update map $\psi$, rather than as a uniform approximation result for the continuous PDE solution $u(z)$?


2. Regarding Theorem 5.3, in Eq. (2) of Appendix C, the authors obtain the inequality $\frac{1}{2}(a+b) \geq \frac{D}{2}$, where $a = \mathbb{E}[\\|\hat u(V)-\psi(V)\\|_2 \mid I]$, $b = \mathbb{E}[\\|\hat u(TV)-\psi(TV)\\|_2 \mid I]$, and $D = \mathbb{E}[\\|\psi(V) - \psi(TV)\\|_2 \mid I]$.

    However, this only implies $\max{a,b}\ge D/2$, and does not directly imply $a\ge D/2$.

    In addition, in the subsequent proof, the authors assume

    $$\mathbb{E}[\\|\psi(V) - \psi(T_{(s, \ell)} V)\\|_2] \ge c a{(s, \ell)}.$$

    But this inequality cannot simply be summed over all $s,\ell$, because the effects of different coordinates may interact with each other and may even partially cancel out. Therefore, stronger assumptions may be needed.

3. Regarding the learned metric, Eq. (2) of the paper uses an axis-aligned diagonal metric and adjusts the relative scales of the spatial and temporal dimensions through learnable parameters $\gamma_x,\gamma_y,\gamma_t$. This design helps learn the trade-off between spatial and temporal scales, but it cannot directly characterize characteristic-aligned transport, where the relevant upstream observations satisfy a relation such as $\Delta x \approx v\Delta t$.

    Have the authors considered using a static flow-aligned metric of the following form?

    $$d(i)= (\Delta x_i-v\Delta t_i)^\top M (\Delta x_i-v\Delta t_i) + \gamma_t^2\Delta t_i^2,$$

    where $v$ and $M\succeq 0$ are global learnable parameters. This design could express characteristic-aligned transport and rotational anisotropy.

---

> ### Author Response · Authors · 2026-07-01
> **Response to Reviewer NuTG**
>
> Thank you for the careful reading.
>
> ### Theorem 5.1 and the missing discretization-error term
>
> We agree with your diagnosis. The decomposition you propose is exactly the distinction needed: the current proof controls the network approximation term for a discrete update rule, but not the discretization error between \\(u\\) and \\(u_h\\).
>
> As detailed in the official comment, we will revise Theorem 5.1 to state approximation of the finite-stencil update map \\(\\psi\\) directly and align the appendix proof with that statement. This is the planned fix: the intended claim is architectural compatibility with local finite-stencil dynamics, not a new convergence theorem for continuous PDE solvers.
>
> ### Theorem 5.3 lower-bound proof
>
> We agree with both issues you identify. The symmetrization step supports a \\(\\max(a,b)\\ge D/2\\) two-world conclusion, not a lower bound for \\(a\\) alone, and the coordinate-wise perturbation inequalities cannot be summed without an additional non-cancellation condition.
>
> As described in the official comment on Theorem 5.3 above, we will revise the theorem as a two-world/minimax non-identifiability statement under a set-level missed-support influence assumption. The proof will then use the valid max lower bound and will no longer derive cumulative missed influence by summing coordinate-wise inequalities.
>
> ### Learned metric and characteristic-aligned transport
>
> A characteristic-aligned metric of the form \\(\\Delta x \\approx v\\Delta t\\) is well motivated for wave-like or advective transport when a meaningful transport velocity \\(v\\) is known or can be reliably estimated.
>
> Our setting is broader. In the real-world atmospheric and pollution datasets, a single velocity satisfying \\(\\Delta x=v\\Delta t\\) is not available: transport can be heterogeneous, variable-dependent, affected by forcing and sources, and only indirectly observed through sparse sensors. The strict characteristic form is therefore most appropriate for wave-like transport in the narrower sense, but less directly applicable to diffusion, mixed advection--diffusion, multivariate atmospheric variables, and real monitoring networks.
>
> The learned \\(\\gamma\\)-scaled metric is intended as a more general and data-adaptive mechanism. It does not explicitly model a tilted characteristic relation \\(\\Delta x-v\\Delta t\\); instead, it learns anisotropic spatial and temporal scaling without requiring a specified velocity field. This makes the same architecture applicable across diffusion-like, transport-like, and heterogeneous real-world regimes. We will clarify this design trade-off in the paper and note characteristic-aligned metrics as a promising specialization when reliable velocity information is available.
>
> ### Unseen-sensor prediction and global reconstruction
>
> FieldFormer is designed to exploit local observational support, so when an entire sensor site is held out, or when evaluation moves far from observed sensors, its locality-aware advantage can diminish. This is exactly why we distinguish sensor-space imputation from sensor-holdout/global reconstruction.
>
> Section 6.1 already states that ImputeFormer is adapted as a fixed-node masked-imputation baseline over the deployed sensor network, rather than as a continuous coordinate-query field model. This matters for Table 4: in the held-out sensor setting, ImputeFormer still operates with a fixed node set and fixed temporal windows, giving it a transductive fixed-topology advantage. It is therefore a strong fixed-node imputer, but it is not solving exactly the same field-estimation problem as coordinate-query baselines.
>
> Table 4 also does not show that Senseiver, or any other global-prior baseline, is consistently best. Senseiver is best only on atmospheric AT RMSE; ImputeFormer is stronger on the remaining AT/RH metrics and on PM10/PM2.5, while SVGP obtains the best \\(V_y\\) wind metrics. This mixed pattern is the main point: once evaluation moves away from local observational support, performance becomes strongly prior-dependent, and no single model reliably solves global reconstruction across all regimes.
>
> FieldFormer performs best in the intended sensor-space regime where local support is available, while fixed-node or global-latent priors can be advantageous in some unseen-sensor settings. This does not contradict our claim; it reinforces our framing that global reconstruction under extreme sparsity is underconstrained and prior-dependent.

---

> > ### Comment · Reviewer_NuTG · 2026-07-09
> >
> > The revised Theorem 5.3 is now logically much cleaner. My remaining question is about Assumption D.4. For which classes of local PDEs or benchmark regimes should we expect the set-level missed-support ambiguity margin to hold? Is this assumption intended as a worst-case/non-identifiability condition rather than a typical-distribution guarantee?
> >
> > Besides, in the sensor-holdout experiments, does ImputeFormer know the full deployed node set, including validation/test sensor identities or coordinates, during training or setup? The current text notes a fixed-node/transductive advantage; a precise statement would help readers interpret Table 4.

---

> ### Author Response · Authors · 2026-07-10
>
> Thank you, we are glad the revised Theorem 5.3 reads cleanly. Both points below are now reflected in the paper.
>
> ### On Assumption D.4 (scope of the ambiguity margin)
>
> It is intended as a **worst-case non-identifiability** condition, not a typical-distribution guarantee. It only asserts the *existence* of one separated, observationally indistinguishable completion, exactly what the two-world/minimax lower bound requires, rather than claiming this ambiguity arises on an average draw. It is non-vacuous when the missed spatial support is non-redundant for the local operator: most cleanly for hyperbolic/advective stencils (upwind transport, SWE), where a finite domain of dependence makes missed upstream points irreducible, and for parabolic stencils (Heat) at the one-step update-map level, where the discrete Laplacian has nonzero coefficients on each spatial neighbor. It degrades (\\(c_{\\mathrm{amb}} \\to 0\\)) when smoothness, low-rank structure, or spatial correlation lets the missed values be recovered from observed coordinates. We have added this as a remark next to the assumption.
>
> ### On ImputeFormer in the sensor-holdout setting
>
> Yes, ImputeFormer is instantiated over the **full** deployed node set and keeps a learnable per-node embedding for every node, including the held-out validation/test sensors. What is withheld is only the held-out *values*: they are masked in the input and excluded from the loss, so no target values leak. It thus imputes at *known nodes of a fixed topology* using node-specific learned parameters, rather than predicting at unseen coordinates like the coordinate-query baselines, so its Table 4 numbers are not strictly comparable to them. We now state this precisely in the ImputeFormer description, the Table 4 caption, the sensor-holdout discussion, and the appendix protocol table.
>
> ### Summary of revisions in this round
>
> 1. Added Remark D.5 after Assumption D.4 framing it as a worst-case/existence condition and characterizing where it binds vs. becomes vacuous (PDE classes; role of \\(c_{\\mathrm{amb}}\\)).
> 2. Extended the Theorem 5.3 remark (Section 5) with a one-line pointer to that scope.
> 3. Clarified ImputeFormer's transductive setup in four places — baseline description and Table 4 caption (Section 6), sensor-holdout discussion, and the appendix protocol table — making explicit that it retains per-node embeddings for held-out sensors while their values are withheld, and noting its holdout results are not strictly comparable to coordinate-query methods.

---

> > ### Comment · Reviewer_NuTG · 2026-07-13
> >
> > Thank you for the detailed clarification. The explanation of Assumption D.4 as an existence-based worst-case/minimax condition, together with the discussion of the PDE regimes in which the ambiguity margin is meaningful or may become vacuous, addresses my concern about its scope. The clarification of ImputeFormer’s fixed-node transductive setup also resolves my question regarding the interpretation and comparability of the sensor-holdout results in Table 4.
> >
> > With these revisions, my main concerns have been satisfactorily addressed. I have no further questions.

---

### Review · Reviewer_hGFy · 2026-06-27

**Summary Of Contributions:**

This paper studies a spatio-temporal field reconstruction problem, and the challenge is learning from sparse sensor dataset to predict the field at unobserved locations. The authors propose a new architecture called FieldFormer, which is a locality-aware transformer designed for sparse sensor networks. In this new algorithm, the learner gathers a fixed-size candidate set of nearby sensor-time tuples for a query point, encodes each neighbor by its relative space-time offset and observed value, scales those offsets by a learnable velocity-scaled (anisotropic) metric, passes the resulting tokens through a small transformer encoder, mean-pools, and maps to the predicted field value with an MLP head. The authors also provide two variants of FieldFormer: FieldFormer-PINN and FieldFormer--Gamma Field. The authors provide two theoretical results, one is about the expressivity of FieldFormer, and the other is about the miss-coverage lower bound. The authors also conduct extensive experiments on three synthetic PDEs and two real-world datasets, and compare FieldFormer with several baselines.

**Audience:**

Yes

**Audience Explanation:**

Yes. Spatio-temporal reconstruction from sparse, persistent sensor networks is of broad interest across the TMLR community.

**Broader Impact Concerns:**

No broader impact statement is needed.

**Claims And Evidence:**

Yes

**Claims Explanation:**

The author gives a strong theoretical guarantee is Theorem 5.1 to support the expressivity of FieldFormer, and Theorem 5.3 to support the miss-coverage lower bound. The author also provides extensive experiments to support the effectiveness of FieldFormer. However, there are some issues with the theoretical results and the empirical results.

1. For the expressivity result, I notice that the proof of Theorem 5.1 only shows that FieldFormer can approximate the one-step explicit finite-difference update map $\psi$ to arbitrary accuracy when the full, exact stencil is supplied. However, it does not show that FieldFormer can uniformly approximate the true continuous solution $u$. Therefore, the expressivity result is only partially supported.

2. For the miss-coverage lower bound, I did not follow the step 3 of the proof, where the author sums over unobserved spatial sites and their time levels. I think this step is not valid, because the conditional risk $R$ is a single nonnegative scalar, and many lower bounds on a single nonnegative scalar combine by max, not sum. Could the author clarify this step?

3. For the empirical results show that FieldFormer outperforms the baselines on the real-world sensor-space imputation tasks, but it does not consistently outperform the baselines on the synthetic sensor-space tasks. Therefore, the claims made in the submission are only partially supported by accurate, convincing and clear evidence.

**Requested Changes:**

Please see the above discussion on the partial support of the theoretical claims, the overstatement on the empirical claims.

---

> ### Author Response · Authors · 2026-07-01
> **Response to Reviewer hGFy**
>
> Thank you for the thoughtful review.
>
> ### Scope of Theorem 5.1
>
> We agree. As detailed in the official comment, the current theorem statement overreaches: the proof supports finite-stencil update-map approximation, not direct uniform approximation of the continuous PDE solution.
>
> We will revise Theorem 5.1 and the appendix proof so the theorem states the finite-stencil expressivity result directly. We will also add the discretization-error caveat separating \\(\\|u-u_h\\|\\) from the FieldFormer approximation term.
>
> ### Miss-coverage lower bound
>
> We agree. As detailed in the official comment on Theorem 5.3, the current proof incorrectly turns a two-world symmetrization argument into a one-world risk bound and also treats coordinate-wise lower bounds as additive.
>
> We will revise the theorem as a two-world/minimax non-identifiability bound and replace the coordinate-wise influence condition with a set-level missed-support condition. This avoids the invalid summation step while preserving the intended dependence on missed spatial influence mass.
>
> ### Empirical claim strength on synthetic sensor-space tasks
>
> Our empirical wording does distinguish "strictly best" from "consistently high-performing." Our intended claim is not that FieldFormer is the top row on every synthetic sensor-space metric, but that it remains consistently comparable to the best baseline across all three synthetic PDE benchmarks.
>
> The results support this interpretation. On Heat, FieldFormer obtains RMSE/MAE of 0.09888/0.07876, compared with the best baseline at 0.09786/0.07786. On Pollution, FieldFormer obtains 0.1154/0.09216, compared with 0.1147/0.09153. On SWE, FieldFormer obtains 0.04644/0.03703, compared with 0.04623/0.03683. Thus, across Heat, Pollution, and SWE, FieldFormer is within roughly one percent of the best method while substantially outperforming several alternatives, especially on SWE.
>
> We will keep this distinction explicit: FieldFormer is consistently competitive on synthetic sensor-space prediction, while its strongest empirical gains appear in the real-world persistent sensor-network setting.

---

### Comment · Action_Editor_z24K · 2026-06-29
**Its time for the discussion period!**

Hi everyone!

We now have all three reviews for this paper. First, I wanted to briefly thank the reviewers for getting them in on time and with great quality. There is a lot of useful feedback for the authors.

I want to remind you, as per TMLRs review procedure, that we are given two weeks for the discussion phase. This includes iterations of author responses and clarifications, possible revisions to the paper (including experiments), and reviewer feedback. For this to be a successful discussion period, I invite both authors and reviewers to be engaged and generous.

As a note to all reviewers, I will not consider your final recommendations unless I see good faith efforts to engage with the authors during this discussion period. We owe them and their work this respect.

Best, Taylor

---

### Author Response · Authors · 2026-07-01
**Response on Theorems 5.1 and 5.3**

## Common response on Theorem 5.1

The reviewers are correct that the current statement of Theorem 5.1 is stronger than what the proof establishes.

The current theorem is written as a uniform approximation statement for the continuous PDE solution:

\\[
\\sup_{z \\in K}\\Vert u(z)-\\hat u(z)\\Vert_2 \\lt \\varepsilon.
\\]

The proof in Appendix C establishes a more specific architectural expressivity claim: given the local stencil values used by an explicit finite-difference discretization, FieldFormer can approximate the corresponding finite-stencil update map

\\[
u_h^{n+1}(\\mathbf{i}) = \\psi(\\{u_h^{n-\\ell}(\\mathbf{i}+\\mathbf{s}) : (\\mathbf{s},\\ell)\\in\\mathcal{S}\\})
\\]

uniformly over a compact set of stencil-value tuples. Thus the proof controls the network approximation error for the discrete local update map, not the discretization error between the continuous PDE solution \\(u\\) and a numerical solution \\(u_h\\).

We will revise Theorem 5.1 to state this result directly. The theorem will be retitled as an expressivity result for **finite-stencil local dynamics** and restated as

\\[
\\sup_{v \\in {\\mathcal V}_{K}} \\Vert \\psi(v)-F_{\\theta}(E(v)) \\Vert_{2} \\lt \\varepsilon.
\\]

Here \\(\\mathcal{V}_K\\) is the compact set of reachable stencil-value tuples and \\(E(v)\\) is the FieldFormer tokenization. We will also revise the appendix proof to end at this finite-stencil approximation bound, deleting the unsupported jump to continuous-solution approximation.

Finally, we will add a remark separating approximation and discretization:

\\[
\\Vert u-\\hat u\\Vert \\le \\Vert u-u_h\\Vert + \\Vert u_h-\\hat u\\Vert.
\\]

The first term requires standard numerical-analysis assumptions such as consistency, stability, regularity, and grid refinement; Theorem 5.1 only addresses the second term. This preserves the intended role of the theorem: architectural compatibility with local finite-stencil dynamics when the relevant local context is available.

## Common response on Theorem 5.3

The reviewers are correct that the current proof of Theorem 5.3 is too strong as written.

First, the symmetrization argument only implies a two-world statement. If \\(V\\) and \\(T V\\) agree on the observed coordinates but induce different outputs, an estimator using only the observed data cannot know which completion is true. The triangle inequality gives a lower bound on the larger of the two risks:

\\[
\\max\\{R(V),R(TV)\\} \\ge \\frac{1}{2}\\Vert \\psi(V)-\\psi(TV)\\Vert_2.
\\]

It does not imply the same lower bound for the original world \\(V\\) alone. We will therefore revise the theorem as a two-world/minimax non-identifiability statement: no estimator can be uniformly accurate over two observationally indistinguishable completions when their updates are separated.

Second, the proof currently sums coordinate-wise influence margins, which is not valid in general because coordinate effects may interact or cancel. We will replace this with a set-level missed-support assumption. For a missed spatial set \\(J\\), let

\\[
A(J)=\\sum_{s \\in J}\\bar a_s.
\\]

The revised assumption states that if \\(J\\) is unobserved, then there exist two observationally indistinguishable stencil completions, differing only on the missed support, whose updates are separated by at least a constant times \\(A(J)\\):

\\[
\\mathbb{E}\\Vert \\psi(V)-\\psi(T_JV)\\Vert_2 \\ge cA(J).
\\]

Under this assumption, the two-world lower bound says that the larger of the two risks is at least a constant times \\(A(J)\\):

\\[
\\max\\{R(V),R(T_JV)\\} \\ge \\frac{c}{2}A(J).
\\]

Averaging over random longitudinal sensor placement with \\(J\\) equal to the missed spatial set gives

\\[
\\mathbb{E}_I[A(I^c)] = 1-\\frac{m_x}{N_x},
\\]

so the original dependence is retained while the theorem is stated as a valid two-world/minimax result.

The revised theorem is therefore not an average-case risk lower bound under the original data distribution alone. It is a conditional non-identifiability result: if sparse sensing leaves missed spatial support that can produce separated but observationally indistinguishable completions, any estimator using only the sensed data must fail on at least one completion.

---

### Author Response · Authors · 2026-07-02
**Revised Manuscript Under Preparation**

We are currently working on the revision based on the feedback, and we will try to upload the revised manuscript by tomorrow. If there are additional comments on the revised version, we will try to address them over the next week.

---

> ### Author Response · Authors · 2026-07-03
>
> We have updated the manuscript along with a summary of revision changes. We look forward to further discussion with the reviewers.